# Cullin5 drives experimental asthma exacerbations by modulating alveolar macrophage antiviral immunity

Haibo Zhang[1,2,3], Keke Xue[1,2,3], Wen Li[1,2,3], Xinyi Yang[1,2,3], Yusen Gou[1,2,3], Xiao Su[4], Feng Qian[1,2,3] ✉ & Lei Sun[1,2,3] ✉

Asthma exacerbations caused by respiratory viral infections are a serious global health problem. Impaired antiviral immunity is thought to contribute to the pathogenesis, but the underlying mechanisms remain understudied. Here using mouse models we find that Cullin5 (CUL5), a key component of Cullin-RING E3 ubiquitin ligase 5, is upregulated and associated with increased neutrophil count and influenza-induced exacerbations of house dust mite-induced asthma. By contrast, CUL5 deficiency mitigates neutrophilic lung inflammation and asthma exacerbations by augmenting IFN-β production. Mechanistically, following thymic stromal lymphopoietin stimulation, CUL5 interacts with O-GlcNAc transferase (OGT) and induces Lys48-linked polyubiquitination of OGT, blocking the effect of OGT on mitochondrial antiviral-signaling protein O-GlcNAcylation and RIG-I signaling activation. Our results thus suggest that, in mouse models, pre-existing allergic injury induces CUL5 expression, impairing antiviral immunity and promoting neutrophilic inflammation for asthma exacerbations. Targeting of the CUL5/IFN-β signaling axis may thereby serve as a possible therapy for treating asthma exacerbations.

Asthma is among the most common chronic inflammatory airway diseases, affecting approximately 300 million individuals worldwide[1,2]. It is characterized by airway hyperresponsiveness (AHR), mucus overproduction, and airway remodeling, causing repeated episodes of wheeze, breathlessness, chest tightness, and cough[3]. Despite significant advances in asthma management, acute asthma exacerbations are not always prevented by standard corticosteroid therapy and account for a major burden in terms of morbidity, mortality, and health care costs[3,4]. Moreover, excessive use of steroids can induce immune suppression and other adverse reactions[5]. Thus, exploring the underlying pathophysiological mechanisms of virus-induced asthma exacerbations and developing novel targeted therapies is necessary.

Asthma is generally a moderate disease characterized by eosinophilia and dominance of T helper 2 (Th2) cells and their hallmark cytokines IL-4, IL-5, and IL-13. However, it can be subject to periods of rapid exacerbations that are mainly provoked by respiratory viral infections such as rhinoviruses, respiratory syncytial virus, influenza viruses, coronaviruses (such as SARS-CoV-2), human metapneumoviruses, bocavirus, and adenoviruses, among others[6,7]. Viral infections can exacerbate asthma through multiple mechanisms. First, viral infection can disrupt epithelial tight junctions, promoting AHR and aggravating asthma[8]. Additionally, viral infections can increase the adherence of pathogenic bacteria to the lung epithelium, potentially contributing to asthma exacerbations[9]. Viral infections can also alter the immune response, further aggravating asthma.

[1]Shanghai Frontiers Science Center of Drug Target Identification and Delivery, School of Pharmaceutical Sciences, Shanghai Jiao Tong University, 200240 Shanghai, P. R. China. [2]National Key Laboratory of Innovative Immunotherapy, Shanghai Jiao Tong University, 200240 Shanghai, P. R. China. [3]Engineering Research Center of Cell & Therapeutic Antibody, Ministry of Education, School of Pharmaceutical Sciences, Shanghai Jiao Tong University, 200240 Shanghai, P. R. China. [4]Unit of Respiratory Infection and Immunity, Shanghai Institute of Immunity and Infection, Chinese Academy of Sciences, 200031 Shanghai, P.R. China. ✉e-mail: fengqian@sjtu.edu.cn; sunlei_vicky@sjtu.edu.cn

For example, the production of neutrophil chemoattractants, such as granulocyte colony-stimulating factor (G-CSF), IL-8, and IL-17, was upregulated in a mouse model of virus-induced asthma exacerbations[10,11]. In addition to the recruitment of eosinophils to the airway, viral infection can increase the number of neutrophils in the airway, which can produce neutrophil extracellular traps, neutrophil elastase (Elane), leukotrienes (LTs), matrix metalloproteinases (MMPs, especially MMP8 and MMP9), and reactive oxygen species, contributing to further eosinophil accumulation and mucus hypersecretion by goblet cells[9,12,13]. Adequate inhaled corticosteroid (ICS) treatment or combination ICS/long-acting β-agonists (LABA) are standard therapies in virus-induced asthma exacerbations. However, these therapies only modestly reduce asthma exacerbation frequency, but not fully prevent immune suppression adverse effects[14,15]. So exploring novel targets for reducing viral load and virus-induced neutrophilic airway inflammation will be key to alleviating virus-induced asthma exacerbations.

Alveolar macrophages (AMs) are the most abundant innate immune cells in the alveoli of the lung, which play an important role in modulating tissue homeostasis, inflammation, injury, and repair[16,17]. During the development of asthma, AMs engage in complex immune regulatory reactions with eosinophils, neutrophils, T lymphocytes, mast cells and epithelial cells[16,18–20]. Besides, the antiviral immunity of AMs plays an important role in regulating respiratory viral infections to inhibit asthma exacerbations[21]. To initiate signaling upon viral infection, AMs detect viral DNA or RNA using a set of pattern recognition receptors (PRRs)[22]. Recognition of viruses by PRRs triggers transduction of downstream signals mainly via adaptor proteins such as mitochondrial antiviral signaling protein (MAVS) or stimulator of interferon genes (STING), which then induce expression of interferons (IFNs) and interferon-associated genes, to inhibit viral replication[23]. Recent evidence has shown that antiviral immunity is deficient in patients with established asthma, which may contribute to viral persistence and the pathogenesis of virus-induced asthma exacerbations[24]. Several studies have shown that the production of antiviral cytokines such as interferon (IFN)-β and IFN-α is lower in patients with asthma or high IgE[24,25]. Others have reported downregulation of Toll-like receptor 7 (TLR7) on AMs of asthma patients, suggesting defects in viral sensing and antiviral immunity[26]. However, the mechanisms underlying inadequate antiviral immunity in virus-induced asthma exacerbations remain unclear.

Cullin5 (CUL5), a member of the Cullin protein family, is a key component of CRL5, which belongs to the largest family of E3 ubiquitin ligases[27]. In the CRL5 complex, CUL5 acts as a molecular scaffold that interacts with the adaptor proteins Elongin B/C, a suppressor of cytokine signaling (SOCS) substrate receptor protein, and a RING protein, RBX1 or RBX2, to transfer ubiquitin for substrate degradation[28]. CUL5 degrades various substrates involved in several biological processes, including cell migration, DNA damage and repair, autophagy, and oncogenesis[29–31]. Furthermore, CUL5 has emerged as a critical regulator of innate immunity and inflammation[30]. Recently, it is reported that CUL5 also plays an essential role in CD4[+] T cell fate choice and allergic inflammation[32]. However, the role of CUL5 in virus-induced asthma exacerbations remains unknown.

In this study, we use an asthma exacerbation mouse model to characterize the mechanisms underlying virus-induced acute asthma exacerbations. Our data show that the expression of CUL5 is increased in alveolar macrophages in the context of the asthmatic microenvironment. Notably, the upregulated CUL5 aggravates asthma exacerbations and mechanistically promotes neutrophil accumulation by inhibiting antiviral immunity and type 1 interferon production. Our findings propose an important function for CUL5 in virus-induced asthma exacerbations, indicating CUL5 as a target for asthma exacerbations treatment.

## Results

### Cullin5 is upregulated in AMs during asthma exacerbations

To determine the mechanism underlying virus-induced asthma exacerbations, we established a murine model of asthma exacerbations by intranasal challenge with influenza virus (H1N1, PR8) in the presence of house dust mite (HDM)-induced allergic airway disease (Fig. 1a). Firstly, methacholine challenge tests (MCTs) were carried out to examine the AHR[33]. As shown in Fig. 1b, in comparison with the control group of mice, the groups treated with HDM presented a higher AHR, including respiratory system resistance (Rrs), respiratory system compliance (Crs), central airway resistance (Rn), tissue damping (G), and tissue elastance (H). Treating the HDM groups with PR8 resulted in a drastic deterioration in AHR compared with HDM induced allergic asthma group or PR8-infected pneumonia group. Then, pathological changes and inflammatory responses in lung tissues were examined. As shown in Fig. 1c and Supplementary Fig. 1a, asthma exacerbation mice exhibited severe mucus deposition, tissue destruction, cell infiltration, and collagen accumulation. The total cell count in the bronchoalveolar lavage fluid (BALF), IgE concentration, and serum albumin (ALB) concentration were significantly increased in HDM + PR8 group compared to other groups (Fig.1d–f). Furthermore, HDM administration resulted in eosinophil infiltration and the production of Th2 cytokines including Il4, Il5, and Il13 (Fig. 1g–i). However, the HDM + PR8 group displayed increased neutrophil infiltration and production of cytokines related to Th1/Th17 and neutrophils including Tnfa, Il1b, Ifng, Il6, Cxcl15, Elane, Il17, Mmp9, and Tgfb (Fig. 1g–i). These results indicate that respiratory influenza infections exacerbate asthma through a switch from Th2-dominated eosinophilic inflammation to a Th1/Th17-dominated neutrophilic response in allergic asthma mice, closely resembling severe neutrophilic asthma in humans.

To identify key regulatory components in this model, we performed RNA-seq analysis using AMs from the PBS, HDM, and HDM + PR8 groups (Supplementary Fig. 1b). We identified 4732 differentially expressed genes (DEGs) in the HDM + PR8 vs. HDM groups (Supplementary data 1). Using Kyoto Encyclopedia of Genes and Genomes (KEGG), we found that besides "asthma," "influenza A," and "Th17 cell differentiation-related signals," the top enriched pathways for these DEGs were also associated with viral infection-related signals including "viral protein interaction with cytokine receptor (CR)," "NOD-like receptor signaling pathway," and "C-type lectin receptor signaling pathway" (Fig. 1j). Therefore, we analyzed these DEGs combined with AMs and viral infection-related genes using the GeneCards database[34]. We discovered that Cul5 was upregulated in the HDM and HDM + PR8 groups and was closely related to the antiviral immunity of AMs (Fig.1k, Supplementary data 2). Given the essential role of CUL5 in regulating the inflammatory response in the lung[30], we selected CUL5 for further studies. Immunofluorescence assays showed that CUL5 was significantly upregulated in macrophages of HDM-treated mice and sustained high expression levels in HDM + PR8 treated mice (Fig. 1l and Supplementary Fig. 1c). Collectively, these results indicate that CUL5 is upregulated in AMs of mice with influenza-induced asthma exacerbations.

### CUL5 deficiency alleviates influenza-induced asthma exacerbations

To explore the potential pathophysiological role of CUL5 in AMs during asthma exacerbations, we generated myeloid-specific Cul5-deficient mice (LysM^Cre Cul5^fl/fl) by breeding Cul5-floxed mice (Cul5^fl/fl) with Lysozyme M-Cre mice (LysM^Cre) (Supplementary Fig. 2). We intranasally treated Cul5^fl/fl and LysM^Cre Cul5^fl/fl mice with HDM with or without influenza virus to induce allergic asthma and asthma exacerbations. In the HDM-induced allergic asthma group, LysM^Cre Cul5^fl/fl mice showed similar changes in mucus deposition, total BALF cell

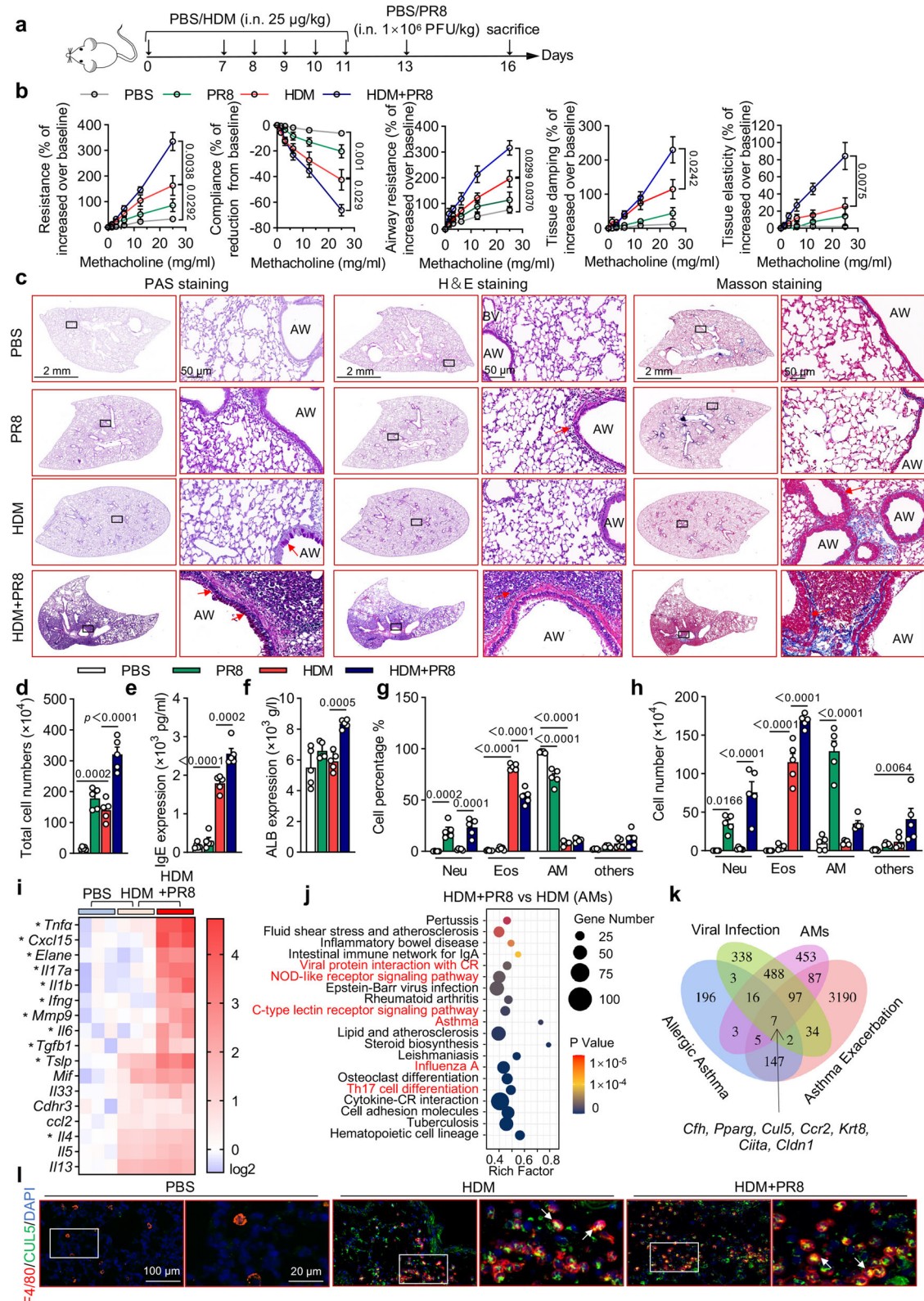

**i**

| | PBS | HDM | HDM+PR8 |
| --- | --- | --- | --- |

* Tnfα
* Cxcl15
* Elane
* Il17a
* Il1b
* Ifng
* Mmp9
* Il6
* Tgfb1
* Tslp
Mif
Il33
Cdhr3
ccl2
* Il4
Il5
Il13

**j** HDM+PR8 vs HDM (AMs)

count, and IgE concentration to *Cul5*^fl/fl mice (Supplementary Fig. 3a–d). Eosinophil infiltration and IL-4, IL-5, and IL-13 expression levels were also similar in HDM-induced allergic asthma LysM^Cre *Cul5*^fl/fl mice and *Cul5*^fl/fl mice (Supplementary Fig. 3e–g). However, in influenza-induced asthma exacerbation mice, LysM^Cre *Cul5*^fl/fl mice showed reduced AHR (Fig. 2a), mucus deposition, tissue destruction, cell infiltration, and collagen accumulation (Fig. 2b and Supplementary

Fig. 3h), total cell count, and M protein (influenza virus ion channel protein) levels in BALF (Fig. 2c–d), and IgE expression (Fig. 2e) compared to *Cul5*^fl/fl mice. As influenza administration combined with the HDM challenges (HDM+PR8) significantly increased neutrophilic inflammation compared to the HDM group (Fig. 1g–i), we assessed the effect of CUL5 on neutrophilic inflammation in the airways. Our data showed that CUL5 deficiency significantly decreased the percentage of

**Fig. 1 | CUL5 is upregulated in alveolar macrophages (AMs) during influenza-induced asthma exacerbations. a** Schemes showing the establishment of PR8-induced pneumonia, HDM-induced allergic asthma and HDM + PR8-induced asthma exacerbations models. **b** Mice were challenged with indicated amounts of methacholine chloride dissolved in PBS. Respiratory system resistance (Resistance) and compliance, airway resistance, and lung tissue damping and elasticity were detected by flexiVent (SCIREQ). **c** Periodic acid-Schiff (PAS), hematoxylin and eosin (H&E), and Masson's trichome-stained lung tissues of the indicated mice. Red arrowheads indicate goblet cells containing mucus (magenta) after PAS staining, inflammatory infiltration after H&E staining, and collagenous fibers after Masson's trichome staining. AW, airway; BV, blood vessel. Scale bars, 2 mm and 50 μm. **d** Total cell counts were determined in the BALF of the indicated mice. **e** ELISA was performed to determine serum IgE levels in the indicated mice. **f** Albumin (ALB) serum levels of mice were determined. **g**–**h** Percentage and number of neutrophil (Neu), eosinophil (Eos), alveolar macrophage (AM), and other cells (others) in BALF. **i** Heatmap summarizing the mRNA expression of asthma exacerbations-related inflammatory factors in the lung tissues of the indicated mice. * Represents $p < 0.05$ in the HDM + PR8 group vs HDM group. **j** Dot graph shows the top 20 KEGG enriched pathways in DEGs between AMs of HDM + PR8-treated mice compared to those of HDM-treated mice, with viral infection-related pathways highlighted in red. Dot size relates to the number of differentially expressed signals. Rich factor represents the ratio of the number of differentially expressed genes (DEGs) located in the pathway entries to the total number of all annotated genes located in the pathway target genes; $p$-value was calculated by one-sided Fisher's exact test. **k** Venn diagram of the overlap between 986 viral infection-related genes (virus, relevance score >10), 1157 AMs-related genes (AMs, relevance score >6) in the GeneCards database, 380 upregulated genes in HDM-induced allergic asthma, and 3572 upregulated genes in HDM + PR8-induced asthma exacerbations. **l** Confocal microscopic imaging of lung tissues of the indicated mice. F4/80, red; CUL5, green; nuclei stained with DAPI, blue. Scale bars, 100 μm and 20 μm. White arrowheads indicate CUL5-expression AMs. Data are representative of three independent experiments (mean ± s.e.m.), $n = 5$ per group per experiment. $p$-values in (**b**) were calculated by one-way ANOVA (Tukey's test) and shown as HDM + PR8 compared with HDM group mice. $p$-values in (**d**)–(**i**) were calculated by one-way ANOVA (Tukey's test). Source data are provided as a Source Data file.

neutrophils (Fig. 2f–g) and the absolute counts of neutrophils, eosinophils, and AMs in the BALF (Fig. 2h). Additionally, the expression of *Il4* and neutrophilic inflammatory cytokines including *Tgfb, Tnfa, Cxcl15, Elane*, and *Mmp9* in asthma exacerbation mice were markedly lower in LysM^Cre *Cul5*^fl/fl mice than in *Cul5*^fl/fl mice (Fig. 2i). In addition, we also detected the expression and role of CUL5 in PR8 induced viral pneumonitis mice model (Supplementary Fig. 4a). Our data showed that although PR8 infection had no effect on the expression of CUL5 in lung tissue (Supplementary Fig. 4b), CUL5 deficient mice still showed a reduction in PR8 induced inflammatory response (Supplementary Fig. 4c–g). Taken together, these results highlight the importance of CUL5 in influenza-induced asthma exacerbations.

Next, we used a double-stranded RNA (dsRNA, Poly(I:C))-induced asthma exacerbations model to further examine the role of CUL5 in asthma exacerbations (Supplementary Fig. 5a). Compared with *Cul5*^fl/fl mice, LysM^Cre *Cul5*^fl/fl mice showed reduced mucus deposition, tissue destruction, cell infiltration, and collagen accumulation (Supplementary Fig. 5b–c). Additionally, CUL5 deficiency reduced capillary rupture, total protein content and total cell count in BALF, and serum IgE levels in mice with dsRNA-induced asthma exacerbations (Supplementary Fig. 5d–g). Flow cytometry showed increased percentage and number of neutrophils in asthma exacerbation *Cul5*^fl/fl mice, which were significantly lower in LysM^Cre *Cul5*^fl/fl mice (Supplementary Fig. 5h–j). Neutrophilic cytokines, including *Tgfb, Tnfa, Cxcl15, Elane*, and *Mmp9*, also showed significantly reduced expression in Poly(I:C)-induced asthma exacerbation mice after CUL5 deficiency (Supplementary Fig. 5k). Collectively, these data demonstrate the essential role of CUL5 in promoting experimental asthma exacerbations, probably by regulating airway neutrophilic inflammation.

## CUL5 promotes asthma exacerbations and neutrophil migration via IFN-β

To determine the mechanism by which CUL5 regulates virus-induced asthma exacerbations, we performed transcriptome RNA-seq analysis to assess the difference in AMs of influenza-induced LysM^Cre *Cul5*^fl/fl and *Cul5*^fl/fl asthma exacerbation mice. Of 748 DEGs, the most enriched pathways were antiviral immunity-related signaling pathways including "NOD-like receptor signaling pathway," "influenza A," "Toll-like receptor signaling pathway" and "viral interaction with CR" by KEGG analysis (Fig. 3a and Supplementary data 3). We also performed RNA-seq analysis of Poly(I:C)-stimulated CUL5-null (*Cul5*^-/-) and wild-type (WT) bone marrow-derived macrophages (BMDMs). Of 1187 DEGs, antiviral immunity-related pathways and the RIG-I-like receptor signaling pathway were also the most enriched (Supplementary Fig. 6a). In addition, among the DEGs, the type I interferon and interferon-associated genes, including *Cxcl10, Ifna14, Ifna15, and Ifnb1*, occupied

prominent positions in the increased expression group, which corresponded with the most enriched pathways (Supplementary Fig. 6b). To further verify the expression levels in the RNA-seq analysis, quantitative reverse transcription PCR (qRT-PCR) was conducted to analyze the mRNA levels of antiviral immunity-related genes. The expression of IFN-β and the interferon-associated genes was significantly increased in CUL5 deficient macrophages of influenza or dsRNA-induced asthma exacerbation mice in vivo, and Poly(I:C)-stimulated BMDMs or PMA-induced THP-1 macrophages in vitro (Fig. 3b, c, Supplementary Fig. 6c–e). These data suggest that CUL5 inhibits antiviral immunity by restricting IFN-β production both in vivo and in vitro.

Previous reports have shown that deficient antiviral immunity may contribute to asthma exacerbations[4,24], and our results also showed that the viral ion channel protein M was decreased in influenza-induced LysM^Cre *Cul5*^fl/fl asthma exacerbation mice (Fig. 2d). We then hypothesized that the relieved asthma exacerbations and neutrophilic inflammation in LysM^Cre *Cul5*^fl/fl mice may due to the enhanced antiviral immunity and IFN-β production. To test this hypothesis, we intranasally treated the influenza-induced asthma exacerbations LysM^Cre *Cul5*^fl/fl mice with an IFN-β neutralizing antibody (anti-IFN-β Ab) (Fig. 3d), which reaggravated mucus deposition, tissue destruction, cell infiltration, and collagen accumulation (Fig. 3e and Supplementary Fig. 6f). The re-elevated total BALF cell counts and serum IgE levels in LysM^Cre *Cul5*^fl/fl mice also verified the asthma exacerbations (Fig. 3f, g). Similarly, flow cytometry showed that the percentages and counts of neutrophils and AMs, and the levels of hallmark cytokines of neutrophilic inflammation were markedly increased in LysM^Cre *Cul5*^fl/fl asthma exacerbation mice administered with IFN-β-neutralizing antibody compared with Isotype controls (Fig. 3h–k). Furthermore, we established a co-culture system to study neutrophil migration in vitro. IL-17A-induced neutrophil migration was significantly attenuated when neutrophils were co-cultured in conditioned medium (CM) from Poly(I:C)-stimulated CUL5-deficient BMDMs (Fig. 3l, m). However, after adding the IFN-β-neutralizing antibody, the decreased neutrophil migration was reversed (Fig. 3m and Supplementary Fig. 6g). Collectively, our data indicate that CUL5 promotes virus-induced asthma exacerbations and neutrophil migration by suppressing IFN-β expression.

## CUL5 negatively regulates antiviral signaling via O-GlcNAc transferase

We then measured the effect of CUL5 on the RIG-I signaling pathway, which is the main pathway affecting type I interferon expression in RNA virus-infected cells[35]. TBK1 and IRF3 showed significantly increased phosphorylation levels in CUL5-null BMDMs or PMA-induced THP-1 macrophages stimulated by dsRNA (Poly(I:C)) and

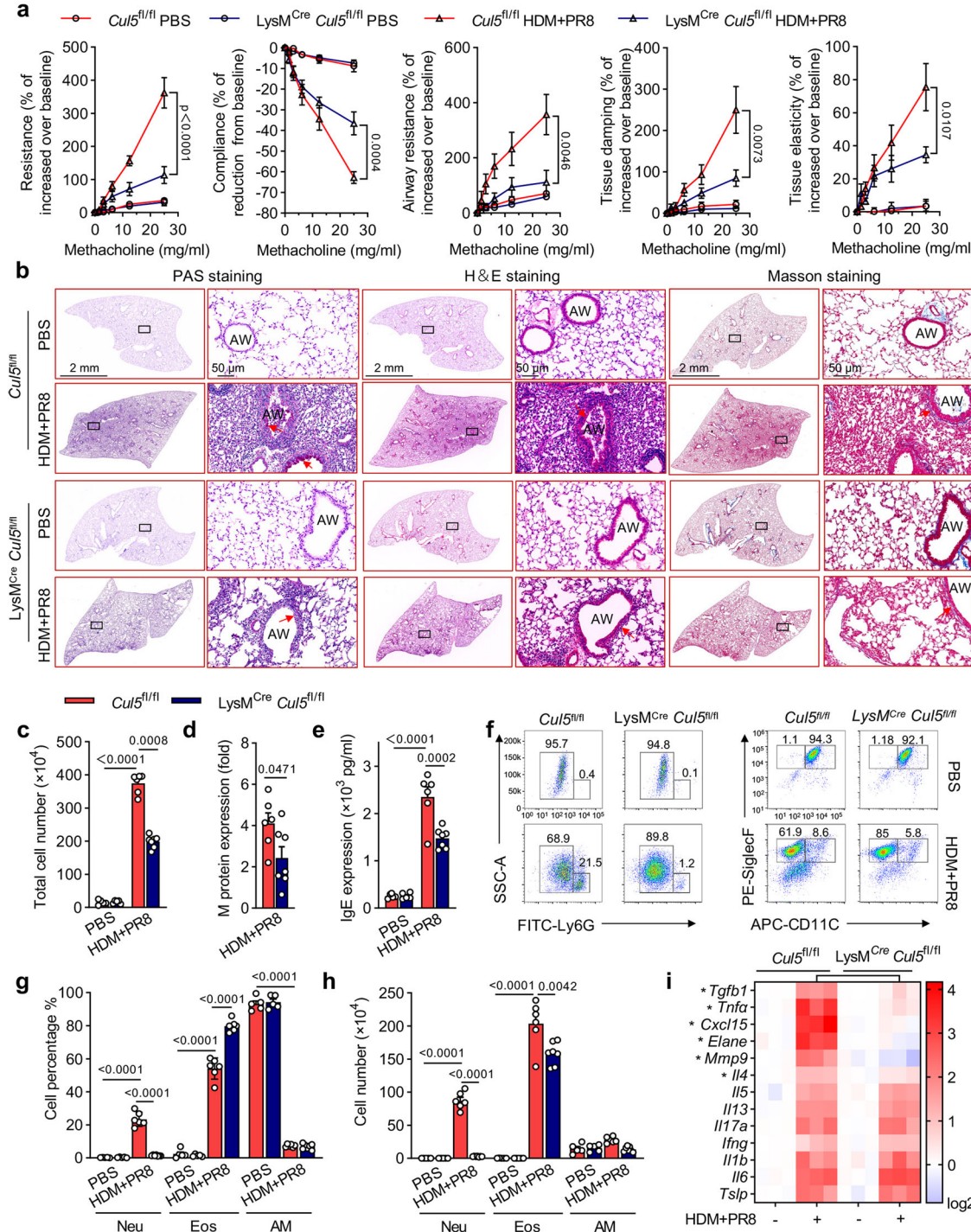

**Fig. 2 | Myeloid-specific CUL5 deficiency alleviates asthma exacerbations.**
**a** Mice were challenged with indicated amounts of methacholine chloride dissolved in PBS. Respiratory system resistance (Resistance) and compliance, airway resistance, and lung tissue damping and elasticity were detected by flexiVent (SCIREQ). $n$ = 5 mice per group. $p$-values shown as $Cul5^{fl/fl}$ treated by HDM + PR8 compared with LysM$^{Cre}$ $Cul5^{fl/fl}$ treated by HDM + PR8 group mice. **b** Periodic acid-Schiff (PAS), hematoxylin and eosin (H&E), and Masson's trichome staining of lung tissues of the indicated mice. Red arrowheads indicate goblet cells containing mucus (magenta) after PAS staining, inflammatory infiltration after H&E staining, and collagenous fibers after Masson's trichome staining. Scale bars, 2 mm and 50 μm. AW, airway.
**c** Total cell counts were determined in the BALF of the indicated mice. **d** Relative influenza virus content in the lung tissues of the indicated groups was quantified based on M protein levels measured by RT-qPCR and shown as ΔCt fold change.

**e** ELISA was performed to detect IgE serum concentration in the indicated mice.
**f** Neutrophil (Neu, Ly6G$^+$), eosinophil (Eos, CD11c$^-$ SiglecF$^+$), and alveolar macrophage (AM, CD11c$^+$ SiglecF$^+$) in BALF were analyzed by flow cytometry.
**g**–**h** Percentages and counts of Neu, Eos, and AM in BALF were determined.
**i** Heatmap summarizing the mRNA expression of *Tgfb, Tnfa, Cxcl15, Elane, Mmp9, Il4, Il5, Il13 Il17, Ifng, Il1b, Il6*, and *Tslp* in the lung tissues of the indicated mice. * Represents $p < 0.05$ in the HDM + PR8-treated LysM$^{Cre}$ $Cul5^{fl/fl}$ group vs the HDM + PR8 $Cul5^{fl/fl}$ group. Data are representative of three independent experiments (mean ± s.e.m.), $n$ = 5, 5, 6, and 7 in the PBS-administered $Cul5^{fl/fl}$, PBS-administered LysM$^{Cre}$ $Cul5^{fl/fl}$, HDM + PR8-administered $Cul5^{fl/fl}$, HDM + PR8-administered LysM$^{Cre}$ $Cul5^{fl/fl}$ groups, respectively. $p$-values in (**d**) were calculated by unpaired two-tailed $t$-test. $p$-values in (**a**), (**c**), (**e**), and (**g**)-(**i**) were calculated by two-way ANOVA (Tukey's test). Source data are provided as a Source Data file.

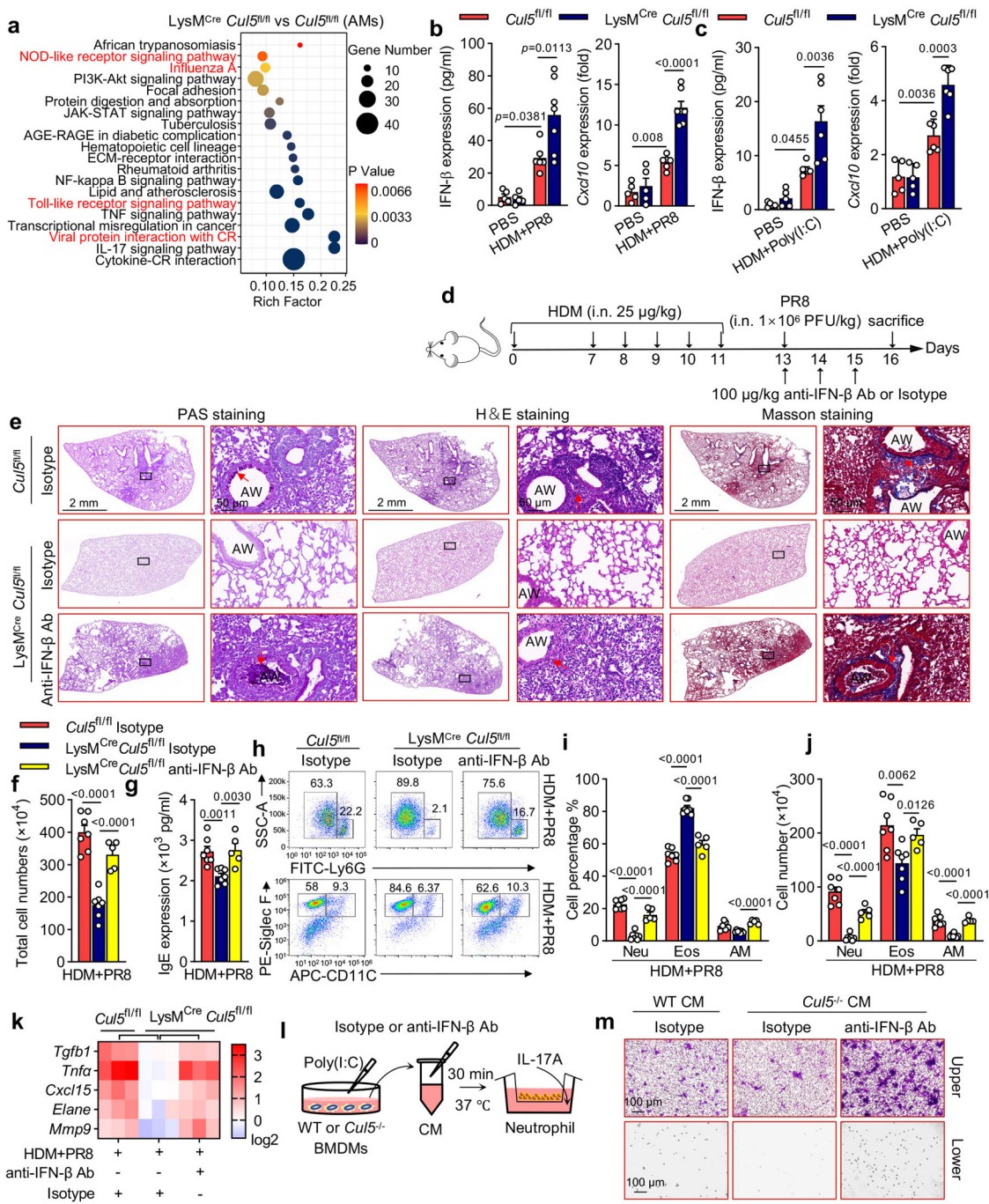

single-stranded RNA (ssRNA) (PolyU) compared to those in WT macrophages (Fig. 4a, b and Supplementary Fig. 7a–d). Moreover, CUL5 deficiency increased the content and proportion of P-IRF3 in the nucleus (Fig. 4c and Supplementary Fig. 7e). To determine the specific target of CUL5 in the RIG-I signaling pathway, we used a dual-luciferase reporter system and observed that CUL5 overexpression markedly decreased the IFN-β luciferase reporter activity induced by RIG-I or mitochondrial antiviral-signaling protein (MAVS), which was increased by CUL5 knockdown (Fig. 4d–e). However, CUL5 did not affect the IFN-β luciferase reporter activity induced by cGAS/STING, TBK1, or IRF3 (Fig. 4d–e), suggesting that CUL5 functions downstream of MAVS and upstream of TBK1. To further examine the effect of CUL5 on MAVS, we measured MAVS expression and activity in the absence of CUL5. We

found that MAVS protein levels were unchanged (Supplementary Fig. 8a). However, CUL5 deficiency promoted MAVS mitochondrial accumulation (Fig. 4f and Supplementary Fig.8b). Additionally, CUL5 overexpression increased MAVS multimerization (Fig. 4g). These results indicate that CUL5 modulates RIG-I signaling by inhibiting the activation, but not the expression, of MAVS.

Given that MAVS activation is regulated by multiple posttranslational modifications (PTMs), such as phosphorylation, ubiquitination, and O-GlcNAcylation[36], we analyzed our transcriptomic RNA-seq data of Poly(I:C)-treated WT and $Cul5^{-/-}$ BMDMs. By analyzing the most enriched biological process (BP) in Gene Ontology (GO), we found that protein glycosylation, especially protein O-linked glycosylation, was upregulated in CUL5 deficient macrophage (Fig. 4h). To verify the

**Fig. 3 | CUL5 promotes asthma exacerbations and neutrophil migration via IFN-β inhibition. a** Dot graph showing the top 20 KEGG enriched pathways in DEGs between alveolar macrophages (AMs) of HDM + PR8-treated LysM^Cre *Cul5*^fl/fl mice and HDM + PR8-treated *Cul5*^fl/fl mice, with antiviral immune response-related pathways highlighted in red. *p*-value was calculated by Fisher's exact test. **b–c** ELISA was performed to determine IFN-β concentration in the BALF and *Cxcl10* expression in the lung tissues of HDM + PR8 and HDM+Poly(I:C)-induced asthma exacerbation mice. *n* = 5, 5, 6, and 7 in the PBS-administered *Cul5*^fl/fl, PBS-administered LysM^Cre *Cul5*^fl/fl, HDM + PR8-administered *Cul5*^fl/fl, and HDM + PR8-administered LysM^Cre *Cul5*^fl/fl groups, respectively, or *n* = 5, 5, 6, and 6 in the PBS-administered *Cul5*^fl/fl, PBS-administered LysM^Cre *Cul5*^fl/fl, HDM+Poly(I:C)-administered *Cul5*^fl/fl, and HDM+Poly(I:C)-administered LysM^Cre *Cul5*^fl/fl groups, respectively. **d** Schemes showing anti-IFN-β antibody treatment in asthma exacerbation mice. **e** Periodic acid-Schiff (PAS), hematoxylin and eosin (H&E), and Masson's trichome-stained lung tissues of the indicated mice. Red arrowheads indicate mucus-containing goblet cells (magenta) after PAS staining, inflammatory infiltration after

H&E staining, and collagenous fibers after Masson's trichome staining. Scale bars, 2 mm and 50 μm. AW, airway. **f** Total cell counts were determined in the BALF of the indicated mice. **g** ELISA was performed to determine serum IgE concentration in the indicated mice. **h** Neutrophil (Neu, Ly6G^+), eosinophil (Eos, CD11c^- SiglecF^+), and AM (CD11c^+ SiglecF^+) in BALF were analyzed by flow cytometry. **i–j** Percentages and counts of Neu, Eos, and AM were determined. **k** Heatmap summarizing *Tgfb*, *Tnfa*, *Cxcl15*, *Elane*, and *Mmp9* mRNA expression in the lung tissues of the indicated mice. Data are representative of three independent experiments (mean ± s.e.m.). *n* = 7, 7, and 5 in the HDM + PR8-administered *Cul5*^fl/fl, HDM + PR8-administered LysM^Cre *Cul5*^fl/fl, and HDM + PR8+anti-IFN-β-administered LysM^Cre *Cul5*^fl/fl groups, respectively. *p*-values were calculated by two-way ANOVA (Tukey's test) (**b–c**), or unpaired two-tailed *t* test (**f–k**). **l** Diagram showing the collection and treatment of conditioned medium (CM). **m** Transwell assay analysis of neutrophil migration ability. Scale bar, 100 μm. Pictures show one out of three biological replicates. Source data are provided as a Source Data file.

glycosylation levels indicated by RNA-Seq, we conducted immunoblotting assays and observed that total protein glycosylation and specific O-GlcNAcylation of MAVS were both increased in CUL5-deficient BMDMs stimulated by Poly(I:C) (Fig. 4i, j and Supplementary Fig. 8c, d). These results suggest that CUL5 modulates the O-GlcNAcylation of MAVS.

As O-GlcNAc transferase (OGT) is the only protein reported to activate MAVS O-GlcNAcylation to date[37], we studied the effect of CUL5 on OGT-induced MAVS O-GlcNAcylation and downstream signal activation. Our data showed that OGT induced MAVS O-GlcNAcylation, which was significantly inhibited by CUL5 overexpression (Fig. 4k). Additionally, in the presence of OSMI-1 (a specific inhibitor of OGT), Poly(I:C)-induced phosphorylation of TBK1 and IRF3 were greatly suppressed in CUL5 deficient macrophages (Fig. 4l and Supplementary Fig. 8e). Similarly, MAVS-induced IFN-β luciferase activity, which was increased by transfection of *CUL5* shRNA, was inhibited by OSMI-1 pretreatment or *OGT* shRNA transfection in a time- and dose-dependent manner (Fig. 4m, n). Furthermore, we discovered that OSMI-1 pretreatment of *Cul5*^-/- BMDMs re-suppressed *Ifnb*, *Cxcl10*, and *Isg15* expression compared to that in WT BMDMs (Fig. 4o). Collectively, these results indicate that CUL5 negatively regulates IFN-β expression by inhibiting OGT-mediated MAVS O-GlcNAcylation.

## CUL5 regulates virus-induced asthma exacerbations via OGT

Given the role of OGT-mediated MAVS O-GlcNAcylation in the CUL5-mediated inhibition of antiviral immunity in vitro, we investigated the role of OGT in CUL5-mediated asthma exacerbations in vivo. Influenza-induced asthma exacerbation mice were intraperitoneally challenged with OSMI-1 (Fig. 5a). IFN-β expression was markedly suppressed, and the expression of viral M protein was further induced in LysM^Cre *Cul5*^fl/fl asthma exacerbation mice injected with OSMI-1 compared to that in LysM^Cre *Cul5*^fl/fl asthma exacerbation mice treated with the control vehicle (Fig. 5b, c). Additionally, LysM^Cre *Cul5*^fl/fl mice injected with OSMI-1 showed signs of aggravated asthma exacerbations, including deteriorated AHR (Fig. 5d), increased mucus deposition, tissue destruction, cell infiltration, and collagen accumulation (Fig. 5e and Supplementary Fig. 8f), and higher total BALF cell counts (Fig. 5f) and serum IgE levels (Fig. 5g), compared with LysM^Cre *Cul5*^fl/fl mice treated with the control vehicle. Moreover, LysM^Cre *Cul5*^fl/fl mice injected with OSMI-1 exhibited higher neutrophil levels and upregulated expression of genes associated with neutrophilic inflammation compared with LysM^Cre *Cul5*^fl/fl mice treated with the vehicle (Fig. 5h–k). These findings demonstrate that CUL5 modulates IFN-β production and virus-induced asthma exacerbations, which is, at least in part, dependent on OGT.

## CUL5 interacts with and promotes OGT degradation

To determine how CUL5 regulates IFN-β production via OGT, we first investigated whether CUL5 interacts with OGT. Using

co-immunoprecipitation (Co-IP), we found that CUL5 interacts with OGT in HEK293T cells (Fig. 6a). CUL5 contains a long stalk-like amino-terminal domain (NTD) comprising three cullin repeats (CR1 to CR3), a signature cullin homology domain (CH), and an C-terminal cullin protein neddylation site on 724[38]. OGT contains a putative PEP-CTERM system TPR-repeat lipoprotein domain (PEP-TPR-lipo, PEP) and a glycosyl transferase family domain (GTF). To map the CUL5 region that interacts with OGT, different truncations of CUL5 or OGT were constructed and transfected into HEK293T cells. The interaction between CUL5 and OGT was blocked in either CUL5 ΔCH + ΔNEDD truncation or OGT ΔGTF truncation (Fig. 6b and Supplementary Fig. 9a). Furthermore, the immunofluorescence assay showed that overexpression of CUL5 (wild type, CUL5 ΔCH, CUL5 ΔNEDD, or CUL5 ΔCR1-3) and OGT (wild type, or OGT ΔPEP) in HeLa cells showed the colocalization of CUL5 and OGT, overexpressed CUL5 ΔCH + ΔNEDD or OGT ΔGTF did not (Fig. 6c and Supplementary Fig 9b). These results indicate that the CH domain and neddylation site is essential for the interaction of CUL5 with OGT through the GTF domain.

As CUL5 is a key component of CRL5, we examined the effect of CUL5 on OGT protein level. Our data showed that CUL5 deficiency greatly increased OGT protein levels in the presence of Poly(I:C) stimulation but did not alter OGT mRNA levels. (Fig. 6d and Supplementary Fig. 10a–c). Additionally, CUL5 overexpression significantly reduced OGT protein levels, which were recovered in the presence of the proteasome inhibitor MG132 but not lysosome inhibitors (Bafilomycin A1 or NH₄Cl) (Fig. 6e). These data suggest that CUL5 induces OGT degradation via the ubiquitin-proteasome pathway.

To further explore the mechanism of CUL5-mediacted ubiquitin-proteasome degradation of OGT, we studied the effect of CUL5 on OGT polyubiquitination in vitro. *CUL5* knockdown by shRNA significantly inhibited OGT ubiquitination, whereas CUL5 overexpression greatly induced OGT ubiquitination (Fig. 6f, g). Furthermore, we found that the overexpression of CUL5 ΔCH + ΔNEDD, which could not bind to OGT, did not affect the ubiquitination of OGT, and had no inhibitory effect on MAVS-induced IFN-β-luciferase activity (Fig. 6g, Supplementary Fig. 11). These results imply the involvement of the CH domain and neddylation site in the ubiquitination of OGT by CUL5. Ubiquitination is mediated by different linkages, including Lys6, Lys11, Lys27, Lys29, Lys33, Lys48, and Lys63[39]. Given that K48-linked polyubiquitin chains mainly target proteins for proteasomal degradation, we transfected HEK293T cells with K48-Ub or K48R-Ub (containing only the 48^th lysine substitution by arginine) plasmids to explore the specific ubiquitin linkage of OGT by CUL5. As shown in Fig. 6h, CUL5 overexpression promoted K48-linked polyubiquitination of OGT but did not alter the ubiquitination level of OGT when K48R-Ub was transfected. These results indicate that CUL5 induces K48-linked polyubiquitination of OGT.

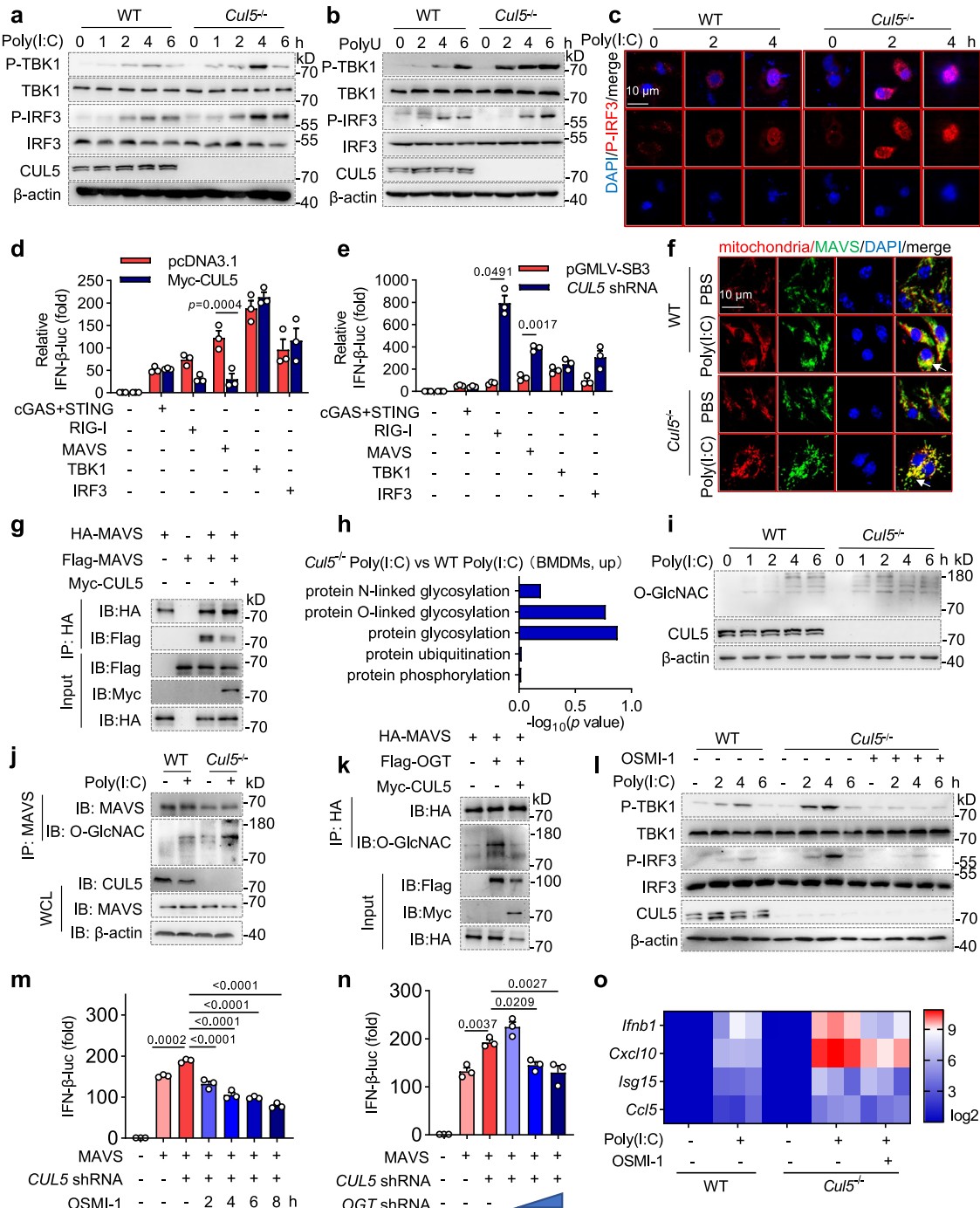

**Fig. 4 | CUL5 inhibits IFN-β expression by suppressing mitochondrial antiviral-signaling protein (MAVS) O-GlcNAcylation in macrophages. a** Immunoblot of phosphorylated (P-)TBK1, TBK1, P-IRF3, IRF3, and β-actin in lysates of BMDMs stimulated by 4 μg/ml Poly(I:C) for the indicated times. **b** Immunoblot of the indicated proteins in lysates of BMDMs stimulated by 4 μg/ml PolyU for the indicated times. **c** Confocal microscopic imaging of P-IRF3 distribution and expression in BMDMs stimulated by 4 μg/ml Poly(I:C) for the indicated times. Scale bars, 10 μm. **d, e** Dual-luciferase reporter system analysis of relative IFN-β-luc activation based on pRL-TK-luc with the indicated co-transfected plasmids. **f** Confocal microscopic imaging of BMDMs treated with 4 μg/ml Poly(I:C) for 4 h. Mitochondria, red; MAVS, green; nuclei stained with DAPI, blue. Yellow in merge indicates MAVS and mitochondria co-localization (white arrowheads). Scale bars, 10 μm. **g** Co-immunoprecipitation (Co-IP) and immunoblotting of HEK293T cells co-transfected with Flag-MAVS, HA-MAVS, and Myc-CUL5. **h** Gene ontology analysis of upregulated posttranslational modifications. X-axis: −log10 (p-value) calculated by Fisher's exact test.

**i** Immunoblot of total protein O-GlcNACylation in lysates of BMDMs stimulated by 4 μg/ml Poly(I:C) for the indicated times. **j** Co-IP and immunoblotting of MAVS O-GlcNACylation in BMDMs treated with 4 μg/ml Poly(I:C) for 4 h. **k** Co-IP and immunoblotting of HEK293T cells co-transfected with HA-MAVS, Flag-OGT, and Myc-CUL5. **l** Immunoblotting of P-TBK1, TBK1, P-IRF3, IRF3, and β-actin in lysates of BMDMs stimulated by 4 μg/ml Poly(I:C) for the indicated times and pretreated with 20 μg/ml OSMI-1 for 30 min. **m, n** Dual-luciferase reporter system analysis of relative IFN-β-luc activation based on pRL-TK-luc with or without OSMI-1 treatment (**m**), or with the indicated co-transfected plasmids (**n**). **o** Heatmap summarizing *Ifnb*, *Cxcl10*, *Isg15*, and *Ccl5* mRNA expression in BMDMs stimulated with 4 μg/ml Poly(I:C) for 6 h. Data in (**a–c, f, g, i–l**) are representative of three independent experiments. Data represent means ± s.e.m. of three biological replicates. *p*-values were calculated by two-way ANOVA (Sidak's test) (**d, e**) or one-way ANOVA (Tukey's test) (**m, o**). Source data are provided as a Source Data file.

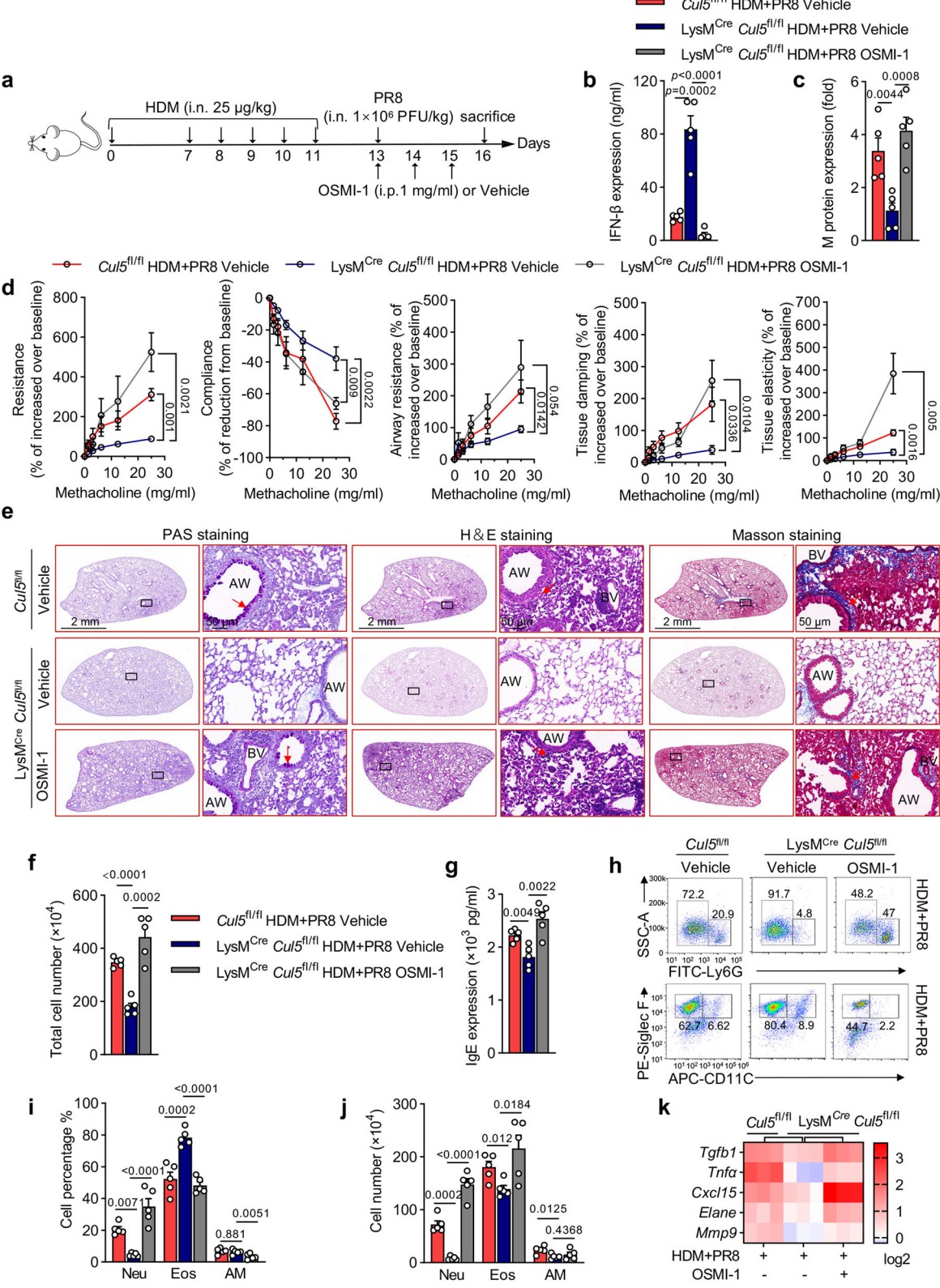

Next, to identify the specific sites of CUL5-mediated ubiquitination in OGT, we predicted potential ubiquitination sites using a bioinformatics tool (Supplementary Table 1)[40], and we predicted several lysine residues (Lys306, Lys459, Lys547, Lys550, Lys996, and Lys999) in OGT (Fig. 6i). Next, we substituted these lysine sites with arginine to establish single-site mutant plasmids (K306R, K459R, K547R, K550R, K996R, and K999R) respectively. Our results indicated

that the K306R, K459R, K547R, and K999R mutations in OGT almost completely abolished CUL5-mediated polyubiquitination (Fig. 6j). These results indicate that CUL5 promotes OGT ubiquitination and degradation by transferring K48-linked ubiquitin to the K306, K459, K547, and K999 sites of OGT.

Being a component of CRL5, CUL5 combines with substrate proteins, depending on the specific substrate receptors (Fig. 6k). The

**Fig. 5 | CUL5 in alveolar macrophages (AMs) regulates virus-induced asthma exacerbations via O-GlcNAc transferase (OGT). a** Schemes of OSMI-1 challenge in the established asthma exacerbation model mice. **b** ELISA was performed to determine IFN-β concentration in the BALF of the indicated mice. **c** Relative influenza virus content in the lung tissues was quantified based on M protein levels measured via RT-qPCR and shown as ΔCt fold change. **d** The airway hyperresponsiveness (AHR) of mice was analyzed by flexiVent. **e** Periodic acid-Schiff (PAS), hematoxylin and eosin (H&E), and Masson's trichrome-stained lung tissues of the indicated mice. Red arrowheads indicate goblet cells containing mucus (magenta) after PAS staining, inflammatory infiltration after H&E staining, and collagenous fibers after Masson's trichome staining. AW, airway; BV, blood vessel. Scale bars, 2 mm and 50 μm. **f** Total cell counts were determined in the BALF of the indicated mice. **g** ELISA was performed to determine serum IgE levels in the indicated mice. **h** Neutrophil (Neu, Ly6G+), eosinophil (Eos, CD11c− SiglecF+), and AM (CD11c+ SiglecF+) in BALF were analyzed by flow cytometry. **i, j** Percentages and counts of Neu, Eos, and AM were determined. **k** Heatmap summarizing the mRNA expression of *Tgfb, Tnfa, Cxcl15, Elane*, and *Mmp9* in the lung tissues of the indicated mice. All mice were treated with HDM + PR8. Data are representative of three independent experiments (mean ± s.e.m.), n = 5 per group per experiment. *p*-values were calculated by unpaired two-tailed *t* test. Source data are provided as a Source Data file.

SOCS family comprises the most extensive substrate receptors for CUL5[27]. To explore the specific substrate receptors for CUL5-mediated OGT ubiquitination, we co-transfected CUL5 and OGT with different members of the SOCS family, including SOCS1, SOCS2, and SOCS3. We observed that SOCS3, but not SOCS1 or SOCS2, promoted CUL5-mediated OGT degradation, which was reversed by MG132 (Fig. 6l and Supplementary Fig. 12a–c). Furthermore, SOCS3 overexpression promoted the interaction between OGT and CUL5 and CUL5-mediated OGT ubiquitination (Fig. 6m, n and Supplementary Fig. 12d). Collectively, these results indicate that CUL5 interacts with and promotes OGT degradation by forming SOCS3 component CRL5 E3 ligases.

### TSLP induces CUL5 inhibitory effect on antiviral immunity

Given the inhibitory effect of CUL5 on antiviral immunity, we attempted to determine the upstream signaling of CUL5 in asthma. Among 3264 DEGs in the HDM vs. PBS groups in the RNA-seq analysis, several viral infection-associated pathways, including "viral myocarditis," "HTLV-1 infection," "Epstein-Barr virus infection," "Herpes simplex virus 1 infection," and "viral protein interaction with CR," were the most enriched in the KEGG analysis (Fig. 7a and Supplementary data 4). Additionally, upon reanalyzing the DEGs in the RNA-seq analysis of OVA or papain-induced allergic asthma in mice in a previous study[18], we found that "Infectious disease: viral" was the most downregulated pathway enriched in the KEGG pathway analysis based on classification at level 2 (Supplementary Fig. 13a, b). These results demonstrate the down-regulation of antiviral immunity in allergic asthma disease. As our data showing that CUL5 was upregulated in HDM-induced allergic asthma mice (Fig. 1k–l), we speculated that the inhibitory effect of CUL5 on antiviral immunity is caused by the allergic asthma microenvironment. Thus, we treated BMDMs with various allergic asthma-related cytokines, including TSLP, IL-33, IL-13, and IL-4. CUL5 expression in macrophages was upregulated by TSLP treatment but was not significantly affected by IL-33, IL-13, or IL-4 treatment (Fig. 7b and Supplementary Fig. 13c, d). Furthermore, OGT protein levels decreased following TSLP stimulation (Fig. 7b and Supplementary Fig. 13c). Immunoprecipitation and immunofluorescence assays showed that the interaction between CUL5 and OGT was promoted by TSLP stimulation in macrophages (Fig. 7c–f). Additionally, TSLP treatment increased OGT ubiquitination in macrophage (Fig. 7g). These results indicate that TSLP promotes CUL5-mediated OGT ubiquitination and degradation.

To further examine the effect of TSLP on CUL5-mediated inhibition of antiviral immunity, we analyzed the activation of RIG-I signal and the expression of interferon-associated genes. Our data showed that the phosphorylation levels of TBK1 and IRF3 were markedly reduced by TSLP pretreatment and were recovered by MLN4924 (CUL5 neddylation inhibitor) treatment (Fig. 7h and Supplementary Fig. 13e). The mRNA levels of *Ifnb* and the interferon-associated genes *Cxcl10* and *Isg15* and the protein levels of IFN-β were also significantly decreased by TSLP treatment, which recovered in the presence of MLN4924 (Fig. 7i, j). Collectively, our results indicate that TSLP in the allergic asthma microenvironment can induce a CUL5-mediated inhibitory effect on antiviral immunity, which may contribute to the exacerbations of asthma.

### IFN-β alleviates virus-induced asthma exacerbations

To evaluate the therapeutic effect of IFN-β on virus-induced asthma exacerbations, we treated asthma exacerbation mice with IFN-β or dexamethasone (Dex) (Fig. 8a). Asthma exacerbation mice showed a remarkable decrease in weight, which was significantly alleviated by IFN-β treatment but not Dex treatment (Fig. 8b). In addition, although the IFN-β treatment and Dex treatment groups both showed reduced respiratory system resistance and airway resistance in virus-induced asthma exacerbation mice, IFN-β treatment was significantly more effective in preserving the respiratory system compliance and tissue damping and elasticity (Fig. 8c), which indicates that IFN-β treatment not only has an effect on the airways, but also has a good effect of improvement on AHR caused by tissue damage. Moreover, our data showed a stronger therapeutic effect of IFN-β than that of Dex in asthma exacerbation mice by reducing histological injury, inflammatory cell infiltration, influenza replication, serum IgE levels, the percentage and count of neutrophils, and the expression of neutrophilic inflammatory genes (Fig. 8d–k and Supplementary Fig. 14). These results suggest that IFN-β could be a better treatment option than corticosteroids for virus-induced asthma exacerbations.

Collectively, our results demonstrate that CUL5, which is upregulated by the epithelial alarmin TSLP in allergic asthma, can contribute to the "insensitive" state of AMs by inhibiting IFN-β production following the viral infection and driving the development of virus-induced asthma exacerbations (Fig. 9).

## Discussion

Asthma exacerbations can be severe and life-threatening and are mainly caused by respiratory viral infections. Defective antiviral immunity is believed to contribute to the development of exacerbations, although the underlying mechanism is not fully understood. Here, we found that during asthma, the epithelial alarmin TSLP induces the expression of CUL5, which acts as a master regulator of IFN-β production in AMs, thereby controlling neutrophil accumulation and its associated effects in the airways. Thus, our study suggests IFN-β and CUL5 as promising therapeutic targets in acute asthma exacerbations.

Since the onset of the COVID-19 pandemic, the impact of viral infections on patients with respiratory system diseases has received increasing attention[41,42]. Respiratory viral infections are the leading cause of asthma exacerbations and cause immunological and morphological changes that promote the development of asthma exacerbations[43]. The conventional treatments for asthma exacerbations are glucocorticoids and beta 2-agonists; other approaches, such as omalizumab (anti-IgE) and viral vaccines, have also been tested clinically[44]. However, these treatments only provide partial improvement in symptoms and show limited effectiveness[45]. Given the regulatory effect of antiviral immunity on airway inflammation, identifying novel immunotherapeutic targets will provide new directions for the treatment of asthma exacerbations. In our study, we observed an upregulation of CUL5 in AMs with virus-induced asthma exacerbations. CUL5-deficient mice presented less severe asthma pathogenesis than control mice. Additionally, CUL5 modulated neutrophilic inflammation in the airways by regulating IFN-β production. IFN-β is considered an effective cytokine against viral infections[46]. The

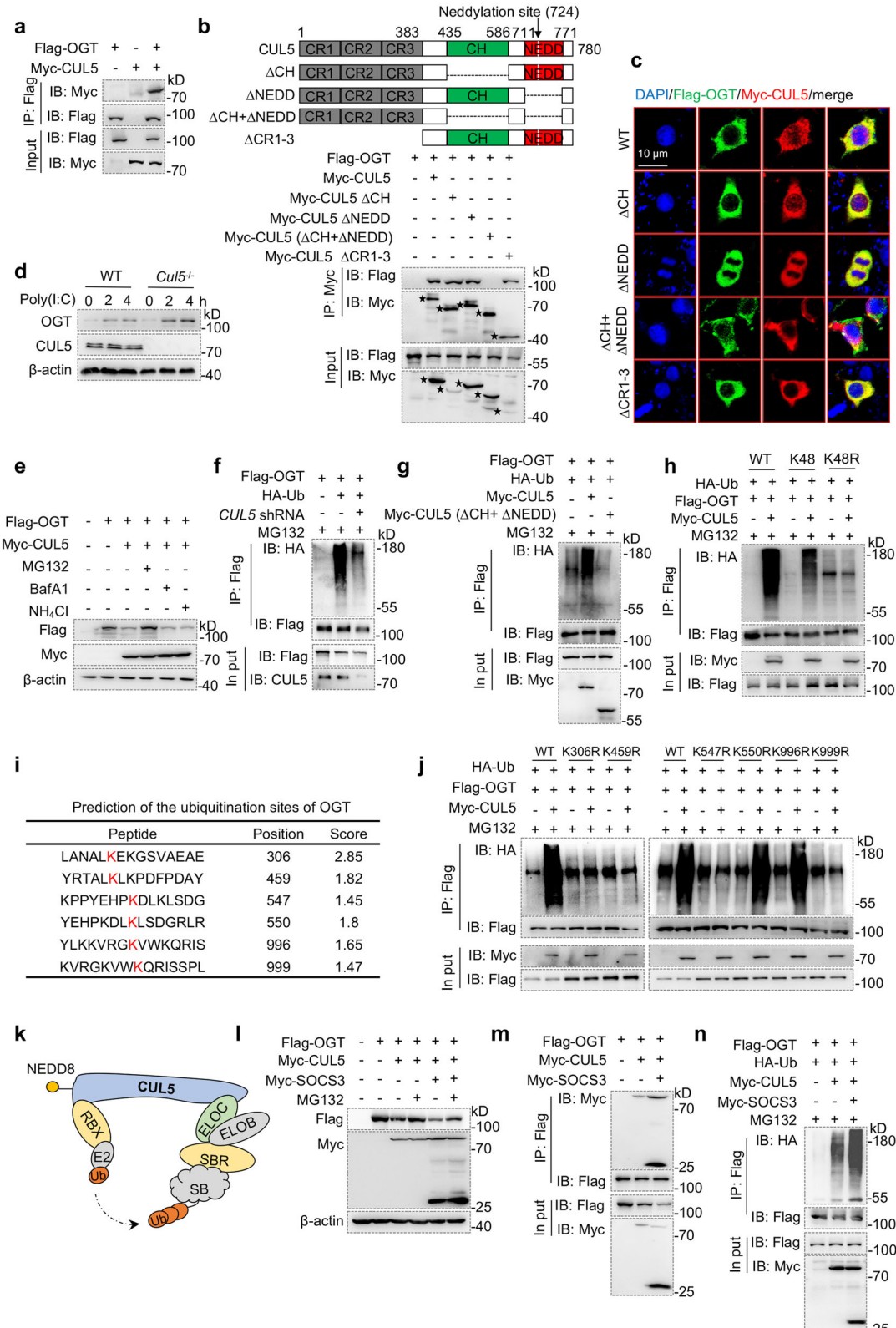

exacerbations of asthma are induced in part by the inhibition of primary antiviral defense caused by reduced expression of IFN-β[2,25,47]. For example, impaired IFN production contributes to MLKL-mediated necroptosis in the lungs and promotes AHR[48]. Moreover, impaired IFN-β production has been shown to activate group 2 innate lymphoid cells (ILC2s), which induces type 2 inflammation in asthma[49]. Based on the effect of IFN-β in modulating asthma exacerbations, over the years,

extensive research has been conducted on IFN-β treatment for virus-induced asthma exacerbations. For example, a recent clinical trial showed that IFN-β treatment within 24 h, once developed a defined cold/flu, significantly enhanced morning peak expiratory flow recovery, reduced the need for additional treatment, and boosted innate immunity in difficult-to-treat asthma patients[50]. Other clinical trials also emphasize the antivirus type I IFN is more sensitive in asthma

**Fig. 6 | CUL5 interacts with and promotes the degradation of O-GlcNAc transferase (OGT). a** Co-immunoprecipitation (Co-IP) and immunoblot analysis of HEK293T cells co-transfected with Flag-OGT and Myc-CUL5. **b** Co-IP and immunoblot analysis of HEK293T cells co-transfected with Flag-OGT, Myc-CUL5, and Myc-CUL5 mutants including CUL5 ΔCH, CUL5 ΔNEDD, CUL5 ΔCH + ΔNEDD, and CUL5 ΔCR1-3. The protein location is marked by black stars. **c** Confocal microscopic imaging of HeLa cells co-transfected with Flag-OGT (green), Myc-CUL5, and Myc-CUL5 mutants (red). Nuclei were stained with DAPI (blue). Yellow in merge indicates the co-localization of Myc-CUL5 and Flag-OGT. Scale bars, 10 μm. Stars represent protein size in the indicated group. **d** Immunoblot analysis of OGT expression in lysates of BMDMs stimulated by 4 μg/ml Poly(I:C) for indicated times. **e** Immunoblot analysis of the indicated protein expression in lysates of HEK293T cells co-transfected with Flag-OGT and Myc-CUL5. Before collecting cell lysis, cells were pretreated with 10 nM MG132, 200 nM Bafilomycin A1 (BafA1), or 4 uM NH$_4$Cl. **f** Immunoblot analysis of OGT ubiquitination in HEK293T cells co-transfected with Flag-OGT, HA-Ub, and *CUL5* shRNA. **g** Immunoblot analysis of OGT ubiquitination in HEK293T cells co-transfected with Flag-OGT, Myc-CUL5, HA-Ub,

and Myc-CUL5 ΔCH + ΔNEDD. **h** Immunoblot analysis of OGT ubiquitination in HEK293T cells co-transfected with Flag-OGT, Myc-CUL5, K48-linked HA-Ub, and the mutant ubiquitin HA-Ub (K48R). **i** The top six scores of potential ubiquitination sites on OGT predicted according to the BDM-PUB online website. **j** Immunoblot analysis of OGT ubiquitination with K306R, K459R, K547R, K550R, K996R, and K999R mutations in HEK293T cells co-transfected with HA-Ub, Flag-OGT, and Myc-CUL5. **k** Structural representation of CRL5 E3 ligases. ELOC, Elongin C; ELOB, Elongin C; SBR, substrate receptor protein; SB, substrates; RBX, RING-box protein; E2, ubiquitin-carrier enzymes; Ub, ubiquitin; NEDD8, Neural precursor cell-expressed developmentally downregulated 8. **l** Immunoblot analysis of the indicated protein expression in lysates of HEK293T cells co-transfected with Flag-OGT, Myc-CUL5, and Myc-SOCS3. **m** Co-IP and immunoblot analysis of HEK293T cells co-transfected with Flag-OGT, Myc-CUL5, and Myc-SOCS3. **n** Immunoblot analysis of OGT ubiquitination in HEK293T cells co-transfected with Flag-OGT, Myc-CUL5, HA-Ub, and Myc-SOCS3. Data are representative of three independent experiments. Source data are provided as a Source Data file.

patients with neutrophilic airway inflammation and in those prescribed high doses of inhaled corticosteroids[24]. These data indicated that the therapeutic effect of IFN-β in asthma exacerbations patient is complicated, and is/probably influenced by the timing of administration and pathological characteristics of subjects. Here, our data showed that CUL5 can suppress IFN-β production and promote neutrophilic inflammation in asthmatic airways. Treating influenza-induced asthma exacerbations mice with IFN-β had a significant therapeutic effect, especially in improving AHR, neutrophilic inflammation, and IgE production. It is evident that our results are based on a murine model of asthma which is different from human. In clinical practice, asthma exacerbations are primarily characterized by bronchi contract, airway swelling, mucus secretion, and a relatively week inflammatory response compared to animal models. However, both in human and mice, virus-induced asthma exacerbations are closely associated with airway inflammation. Patients with asthma exacerbations show a significant infiltration of neutrophils in the airway, the airway swelling and mucus secretion, which are all due to airway inflammation[24]. Although human airway macrophages, including macrophages located on the luminal surface of the alveolar space and macrophages associated with the epithelium of the conducting airways, have some phenotypic differences from mouse alveolar macrophages, they also participate in regulating airway inflammation in asthma exacerbations. Therefore, we believe that unveiling the effect of CUL5 on alveolar macrophages in mice model can also provide some reference value for further determining the role of CUL5 in human. Therefore, our data indicates that CUL5 is a potential therapeutic target for virus-induced asthma exacerbations, and further research is required to validate its role in human and investigate the clinical application of IFN-β in treating asthma exacerbations.

GlcNAcylation is a PTM that adds O-linked β-N-acetylglucosamine (O-GlcNAc) to serine or threonine residues of various proteins. This protein modification interacts with key cellular pathways involved in transcription, translation, and proteostasis;[51] hence, dysfunction of O-GlcNAcylation contributes to several diseases such as cancer, diabetes, neurodegenerative and cardiovascular diseases, and viral infections[52,53]. In eukaryotes, only two conserved enzymes are involved in this process. OGT adds uridine diphosphate N-acetylglucosamine (UDP-GlcNAc) to proteins, whereas O-GlcNAcase (OGA) removes them[51]. As UDP-GlcNAc is generated from the hexosamine biosynthetic pathway, which is pivotal for the cellular metabolism of amino acids, lipids, and nucleotides, the level of O-GlcNAcylation is mainly influenced by the availability of UDP-GlcNAc and the downstream metabolites of glucose[51]. Recently, a study on 3T3-L1 adipocytes showed that fluctuations in UDP-GlcNAc concentrations may not fully reflect changes in extracellular glucose amounts[54], which indicates that there may be other cellular mechanisms for regulating O-GlcNAcylation,

such as directly regulating the expression or activity of OGT. Several reports have shown that OGT activation is regulated by PTMs[55,56]. For example, OGT can be phosphorylated at different sites. GSK3β phosphorylates OGT at Ser3 and/or Ser4 to enhance OGT activity[57], and checkpoint kinase 1 phosphorylates OGT at Ser20, which appears to improve its stability[58]. Here, we found that CUL5 regulates antiviral immunity by promoting the ubiquitination and degradation of OGT, which then induces the O-GlcNAcylation and ubiquitination of MAVS for downstream antiviral signaling activation. Thus, these findings not only shed light on the mechanism by which CUL5 regulates antiviral immunity but also identified another form of PTMs in OGT. Our study indicates that examining the crosstalk between different PTMs could help us further understand the regulation of O-GlcNAcylation and offer more therapeutic targets for O-GlcNAcylation in diseases.

Several studies have shown that viral infections cause asthma exacerbations; however, the interaction between pre-existing asthmatic inflammation and viral infections has rarely been investigated. Recently, it has been proven that pre-existing asthmatic inflammation in the lower airways may modify the immune response to viral infection, leading to delayed viral clearance, persistent virus-induced inflammation, and amplification of asthma exacerbations[59]. Researchers have measured the levels of IgE to establish whether a relationship exists between IgE levels and the likelihood of virus-induced exacerbations. They found that the odds ratio for virus-induced wheezing in children over 2 years of age attending the emergency department was 4.4. If rhinovirus was detected, the concomitant presence of specific IgE increased the odds ratio to 17[60,61]. In our study, CUL5 expression was upregulated by TSLP during allergic inflammation, and CUL5 upregulation did not alter airway inflammation upon exposure to HDM alone. However, CUL5 upregulation significantly promoted lung injury and modulated the immune response in terms of cytokine and chemokine induction by both antigen exposure and virus infection. Thus, our results suggest that early exposure to allergens may contribute to deficiencies in antiviral immunity in asthma exacerbations. Therefore, it is believed that both early allergen exposure and late viral infection play important roles in asthma exacerbations pathogenesis, and targeting the two stages in combination may provide an additional therapeutic approach to the treatment of asthma exacerbations.

In summary, our data demonstrate that CUL5 is a modulator of virus-induced asthma exacerbations. The epithelial alarmin TSLP induces the expression of CUL5, which then interacts with and degrades OGT via K48-linked ubiquitination, inhibiting MAVS O-GlcNAcylation and IFN-β production, and eventually inducing asthma exacerbations. Our findings provide insight into the underlying mechanisms and treatment targets in virus-induced asthma exacerbations.

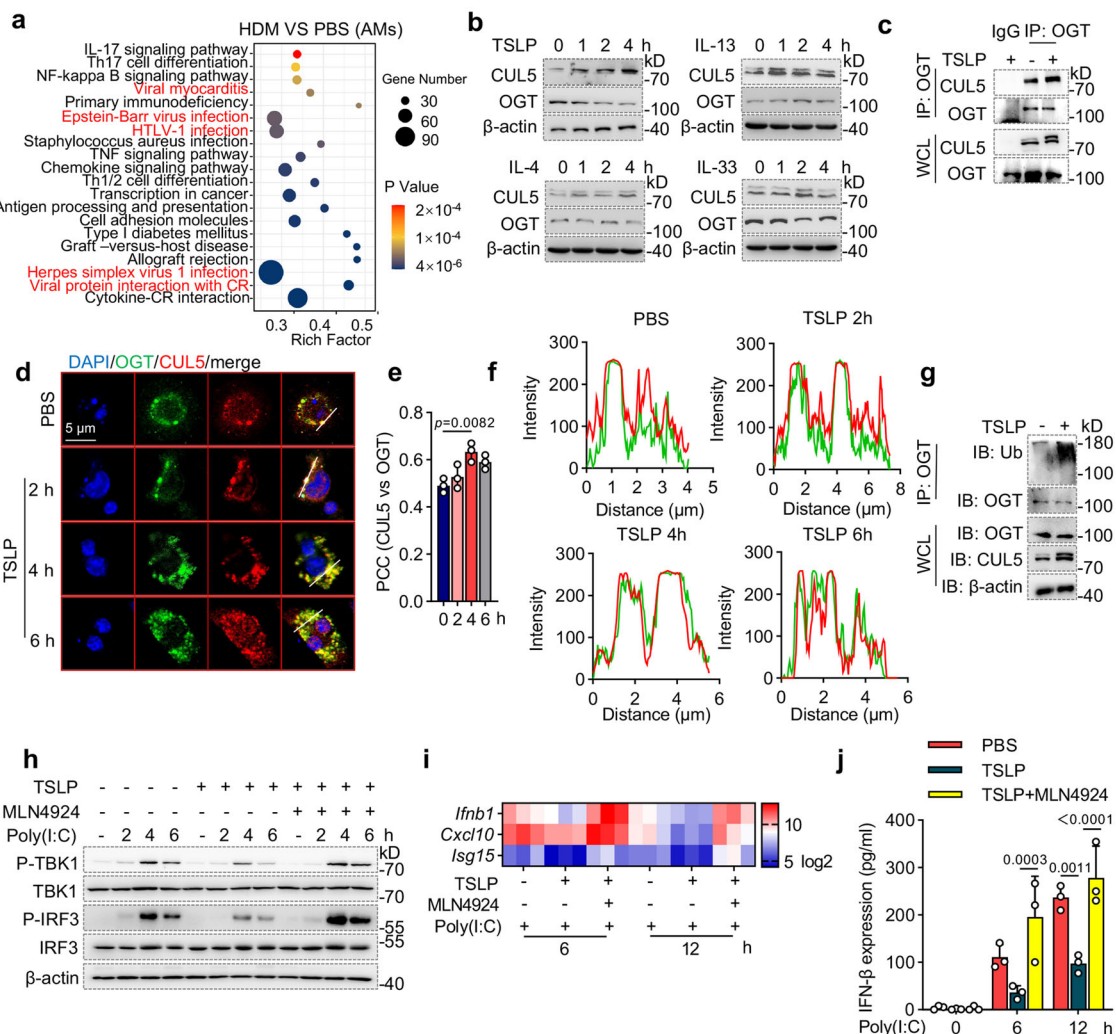

**Fig. 7 | Thymic stromal lymphopoietin (TSLP) induces CUL5-mediated inhibition of antiviral immunity. a** Dot graph showing the top 20 enriched pathways in the alveolar macrophages (AMs) of HDM-treated mice compared with those in PBS-treated mice, with viral infection-related pathways highlighted in red. Dot size represents the number of differentially expressed signals. Rich factor represents the ratio of the number of differentially expressed genes (DEGs) located in the pathway entries to the total number of all annotated genes located in the pathway target genes. *p*-value was calculated by Fisher's exact test. **b** Immunoblotting of O-GlcNAc transferase (OGT) and CUL5 expression in lysates of bone marrow-derived macrophages (BMDMs) stimulated by 20 μM TSLP, 20 μM IL-33, 20 μM IL-4, and 100 μM IL-33 for indicated times. **c** Co-immunoprecipitation (Co-IP) and immunoblotting of BMDMs treated by 20 μM TSLP for 4 h. **d** Confocal microscopic imaging of BMDMs treated by 20 μM TSLP for indicated times. CUL5, red; OGT, green. Nuclei were stained with DAPI (blue). Yellow in merge indicates the co-localization of CUL5 and OGT. Scale bars, 5 μm. **e** Pearson's correlation coefficient

(PCC) was determined to evaluate the correlation of voxel intensity between the CUL5 (red) and OGT (green) channels in (**d**). **f** The fluorescence intensity on the white lines in (**d**) was calculated. **g** Immunoblot analysis of OGT ubiquitination in BMDMs treated with 20 μM TSLP for 4 h. **h** Immunoblot analysis of the indicated proteins in lysates of BMDMs stimulated by 4 μg/ml Poly(I:C) and pretreated with 20 μM TSLP with/without 10 μM MLN4924. **i** Heatmap summarizing *Ifnb, Cxcl10*, and *Isg15* mRNA expression in BMDMs stimulated by 4 μg/ml Poly(I:C) and pretreated with 20 μM TSLP with/without 10 μM MLN4924. **j** ELISA was performed to determine IFN-β expression in BMDMs stimulated by 4 μg/ml Poly(I:C) and pretreated with 20 μM TSLP with/without 10 μM MLN4924. Data in (**b**)–(**d**) and (**g, h**) are representative of three independent experiments. Data represent the means ± s.e.m., *n* = 3 biological replicates. *p*-values were calculated by one-way ANOVA (Tukey's test) in (**e**), or two-way ANOVA (Tukey's test) in (**j**). Source data are provided as a Source Data file.

## Methods
### Mice
6-8 weeks C57BL/6 WT mice were obtained from the SLAC Laboratory Animal Corporation (Shanghai, China). *Cul5*^fl/fl^ mice on C57BL/6 background were developed by Biocytogen Pharmaceuticals (Beijing, China) through conditionally containing a *loxP* sequence flanking the fourth and fifth exons of *Cul5*. *Cul5*^fl/fl^ mice hybridized with LysM^Cre^ mice (Jackson Laboratory, Bar Harbor, ME, Strain #004781) to obtain LysM^Cre^ *Cul5*^fl/fl^ mice. All mice were housed (4–5 animals per cage) at 18–23 °C with 40–60% humidity under a 12/12 h light/dark cycle with ad libitum access to food and water and keep in SPF environment. The housing, breeding, and experimental protocols complied with the

National Institutes of Health Guide for the Care and Use of Laboratory Animals, and were approved by the Biological Research Ethics Committee of Shanghai Jiao Tong University (approval number A2018075).

### Reagents
PR8 (PR8A/Puerto Rico/8/34 (H1N1)) virus was kindly provided by Professor Xiao Su, Shanghai Institute of Immunity and Infection, Chinese Academy of Sciences. Recombinant mouse IL-4 (mIL-4, 404-ML-050), mIL-13 (413-ML-050), mTSLP (555-TS-010/CF), mIL-33 (3626-ML-010), mM-CSF (416-ML-050/CF), and mIL-17A (421-ML-025) and ELISA kits for mIFN-β (DY8234-05), mIL-4 (DY404-05), mIL-5 (DY405-05), and mIL-13 (DY413-05) were purchased from R&D Systems (Minnesota,

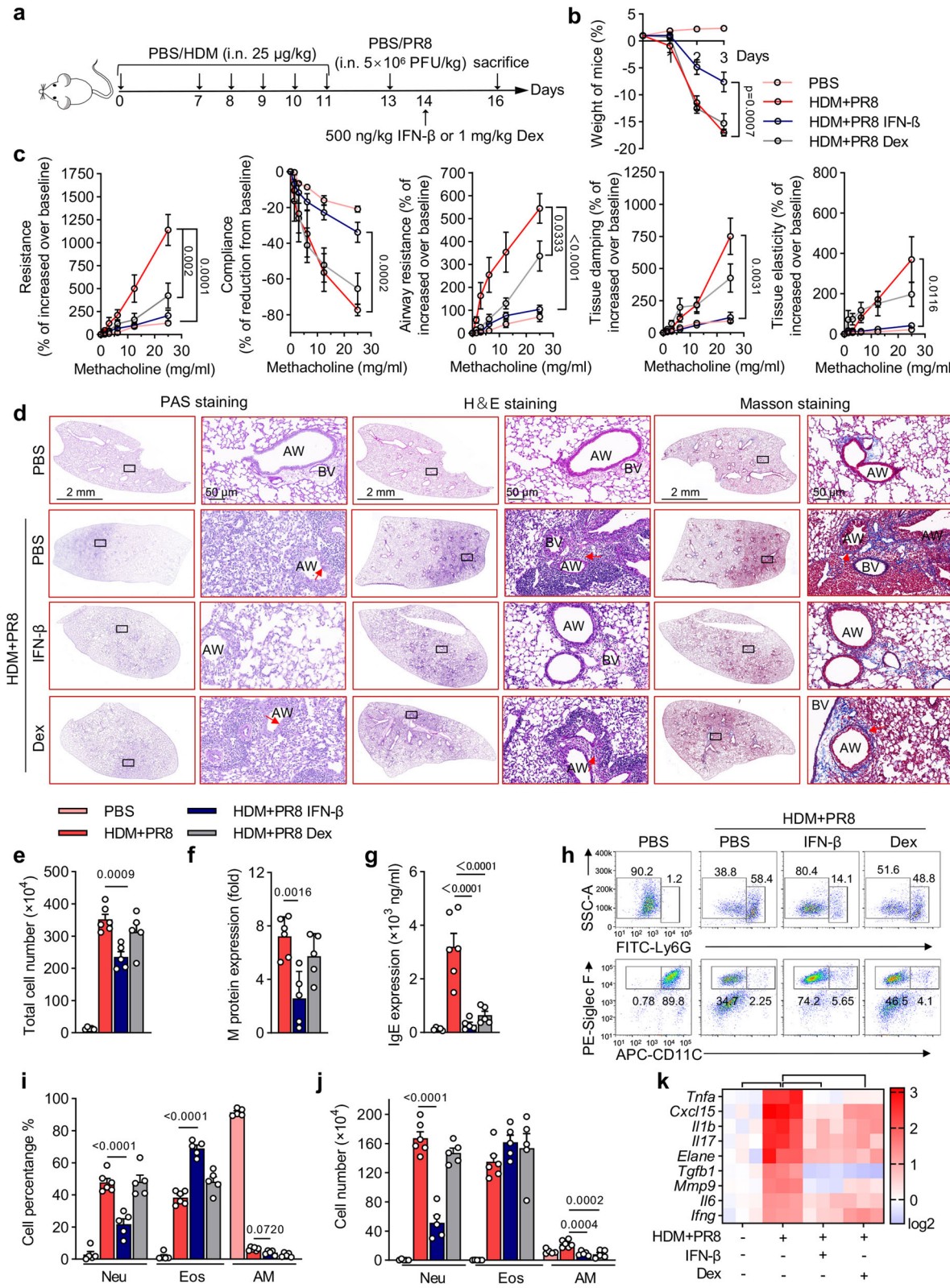

USA). Antibodies in this study are listed in Supplementary Table 2. OSMI-1 (1681056-61-0) and polyethylene glycol 300 (PEG300, 25322-68-3) were purchased from MedChemExpress (Monmouth Junction, NJ, USA). And the following reagents were used: acetylcholine chloride (Shanghai Yuanye Bio-Technology Co., Ltd., 62-51-1), Poly(I:C) (InvivoGen, tlrl-pic), ssPolyU naked (InvivoGen, tlrl-sspu), Lipofectamine 2000 transfection reagent (Thermo Fisher Scientific, 11668019),

opti-MEM (Gibco, 31985070), and Luciferase Reporter Assay System (Promega, E1910).

**Flow cytometry**

Isolated cells were first stained using the Zombie UV Fixable Viability Kit (67-68-5, BioLegend, San Diego, CA, USA) and Fc-blocking anti-mouse CD16/32 antibody (156603) to exclude dead cells and non-

**Fig. 8 | IFN-β treatment induces remission of neutrophilic asthma exacerbations. a** Dosage regimens of IFN-β and dexamethasone (Dex) administered to asthma exacerbation mice induced by HDM and PR8. **b** Weights of mice in the indicated groups. **c** Mice were challenged by indicated amounts of methacholine chloride. AHR were evaluated by flexiVent (SCIREQ). **d** Periodic acid-Schiff (PAS), hematoxylin and eosin (H&E), and Masson's trichome-stained lung tissues of the indicated mice. Red arrowheads indicate goblet cells containing mucus (magenta) after PAS staining, inflammatory infiltration after H&E staining, and collagenous fibers after Masson's trichome staining. AW, airway; BV, blood vessel. Scale bars, 2 mm and 50 μm. **e** Total cell counts were determined in the BALF of the indicated mice. **f** Relative influenza virus content in the lung tissues of the indicated groups

was quantified based on M protein levels measured with RT-qPCR and shown as ΔCt fold change. **g** ELISA was performed to determine serum IgE levels in the indicated mice. **h**, Neutrophil (Neu, Ly6G⁺), eosinophil (Eos, CD11c⁻ SiglecF⁺), and alveolar macrophage (AM, CD11c⁺ SiglecF⁺) in BALF were analyzed by flow cytometry. **i, j** Percentages and counts of Neu, Eos, and AM in BALF were determined. **k** Heatmap summarizing the mRNA expression of *Tnfα, Cxcl15, Ifnb, Il17, Elane, Tgfb Mmp9, Il6,* and *Ifng* in the lung tissues from the indicated mice. Data represent the mean ± s.e.m. of one out of three independent experiments. $n$ = 5, 6, 5, and 5 in the PBS, HDM + PR8, HDM + PR8 IFN-β, and HDM + PR8 Dex groups, respectively. $p$-values were calculated by one-way ANOVA (Tukey's test). Source data are provided as a Source Data file.

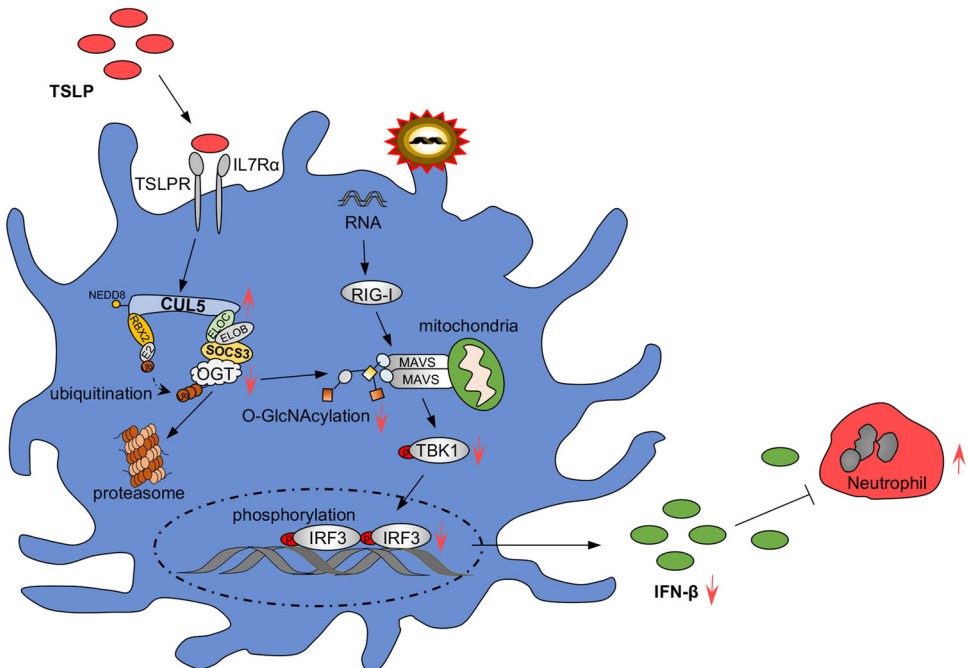

**Fig. 9 | Schematic diagram for the role of CUL5 in virus-induced asthma exacerbations.** CUL5 modulates the antiviral immunity of AMs and induces asthma exacerbations by inhibiting IFN-β production, leading to neutrophil recruitment and accumulation in the airways. Thymic stromal lymphopoietin (TSLP) induces

CUL5 accumulation in AMs, and CUL5 directly binds to and destabilizes O-GlcNAc transferase (OGT), thereby restricting mitochondrial antiviral-signaling protein (MAVS) O-GlcNAcylation and IFN-β production. Red arrowheads mean the change of indicated proteins/ posttranslational modifications (PTMs).

specific binding. Then, cells were strained with flow antibodies for 45 min at 4°C. Data were recorded using flow cytometry (LSRFortessa, Becton, Dickinson and Company, USA) and analyzed using the FlowJo software (version 10). Each cell lineage was identified as follows: AMs, CD11c⁺Siglec-F⁺; neutrophils (Neu), Ly6G⁺; and eosinophils (Eos), CD11c⁻Siglec-F⁺.

### RNA-seq and data processing

Total RNA was isolated using TRIzol reagent (Invitrogen, Carlsbad, California). RNA-seq detection and data processing were performed by the Shanghai Applied Protein Technology Corporation (Shanghai, China). For data quality control, genes were pre-filtered by counts >20 and reads per kilobase million (RPKM) > 1 in at least one sample. Differentially expressed genes were identified using DESeq2 with a fold change >2 and an adjusted P value (FDR) < 0.01, and the P value was calculated using the Z test (Pearson's chi-square test) in the Agilent GeneSpring GX suite. The statistical significance of the expression of each gene was calculated using the negative logarithm (−log10) of the P value. Advanced volcano plots and KEGG enrichment scatter plots were generated using the OmicStudio tools at https://www.omicstudio.cn/tool, and were analyzed with OmicStudioClassic (1.22.0), OmicStudioKits (2.12.0), and ggplot2 (3.3.3) using R version 4.1.3. According to function classification, KEGG is usually divided into

three levels, level 1, level 2, and level 3. Level 1 contains six categories: metabolism, genetic information processing, environmental information processing, cellular processes, organismal systems, and human diseases (specific species comments may have been cut); level 2 contains approximately 44 categories, including cell growth and death, transcription, and development. Level 3 is for conventional enrichment using hundreds of pathways, namely from level 1 to level 3, that function more specifically. The KEGG enrichment in Figs. 1j, 3a, 7a, and Supplementary Fig. 7a was analyzed at level 3. The KEGG enrichment in Supplementary Fig. 13a, b was analyzed at level 2.

### Animal models

House Dust Mite extract containing *Dermatophagoides pteromyssimus* were purchased from Greer Laboratories Inc (Cat. No. XPB82D3A25, Lot. No. 385931). The protein lyophilized powder was dissolved into a stock solution of 5 mg/ml, and aliquoted to store in −80 °C according to the instructions. For the virus-induced asthma exacerbations model, six- to eight-week-old mice (male and female) were randomly divided into control and model groups. Then mice were anesthetized with 150 μl 5% (m/v) chloral hydrate via intraperitoneal injection and sensitized intranasally using 250 μg/kg HDM in 20 μl PBS, 10 μl per nostril on day 0. On days 7−11, mice were challenged daily with 250 μg/kg HDM in 20 μl PBS intranasally. On day 13, mice were intranasally

administered PR8 ($1×10^6$ PFU/kg) or intratracheally administered Poly(I:C) (10 mg/kg). In the anti-IFN-β-treated group, LysM$^{Cre}$ $Cul5^{fl/fl}$ mice were intranasally treated with 20 μl 100 μg/kg anti-IFN-β antibody (R&D Systems, clone 1176D, MAB8234) on days 13–15. The isotype groups were treated with an equal volume of normal rabbit IgG control (R&D Systems, clone 60024 B, MAB1050). In the OSMI-1-treated group, OSMI-1 was dissolved in vehicle solvent (volume ratio of PEG300: PBS = 3: 7), and LysM$^{Cre}$ $Cul5^{fl/fl}$ mice were treated intraperitoneally with 200 μl 1 mg/kg OSMI-1 6 h before PR8 infection, followed by once-daily administration on days 14–15. The vehicle groups were treated with an equal volume of the vehicle. In the mIFN-β or dexamethasone (Dex)-treated group, mice were intranasally treated with 20 μl 500 ng/kg mIFN-β1 (R&D Systems, 8234-MB-010) or 20 μl 1 mg/kg dexamethasone (MCE, 50-02-2) on day 14. All asthma-aggravated mice were euthanized on day 16, and peripheral blood, BALF, and lung tissues were collected for further studies. Each group included at least five mice for each experiment, and the control mice were only treated with sterile PBS without HDM, anti-IFN-β, mIFN-β, or dexamethasone.

## Airway hyperresponsiveness

Airway hyperresponsiveness (AHR) was tested in all mice on day 16 using flexiVent FX equipped with module 2 (flexiVent FX, SCIREQ Scientific Respiratory Equipment Inc., Montréal, Canada). Briefly, the mice were anesthetized intraperitoneally using chloral hydrate. After endotracheal intubation, a very short two-second broadband perturbation followed by forced oscillation perturbation (*SnapShot-150* and *Quick Prime-3*) was performed. Perturbation was performed every 10 s for 3 min following the administration of methacholine chloride (Sigma, 62-51-1). Two deep lung inflations (to 25 cm H$_2$O) were performed at the start of the flexiVent protocol for lung volume history normalization. Then all mice were treated with aerosolized PBS (baseline) and 25 mg/ml, 12.5 mg/ml, 6.25 mg/ml, 3.125 mg/ml, or 1.5 mg/ml methacholine chloride (20 ul per time) using an ANP-1100 Aeroneb Lab Nebulizer unit with small particle size (approximately 2.5–4 microns), with the inspiratory arm of the nebulizer directed through the aerosolization chamber during nebulization (10 s). Detailed lung function parameters were recorded using the forced oscillation method (FOT). Among these parameters, respiratory system resistance (Rrs), and respiratory system compliance (Crs) were measured via single-frequency FOT (*SnapShot-150*). Central airway resistance (Rn), tissue damping (G), and tissue elastance (H) were measured via broadband FOT (*Quick Prime-3*)[62]. The significance and difference of measuring these parameters are as follows: Respiratory system resistance (Rrs) reflects the dynamic resistance in the lung, central airway resistance (Rn) represents the resistance of the central airways, respiratory system compliance (Crs) provides a characterization of the overall elastic properties that the respiratory system needs to overcome during tidal breathing to move air in and out of the lungs, tissue damping (G) reflects the energy dissipation in the alveoli, and tissue elastance (H) reflects the energy conservation in the alveoli.

## Albumin (ALB) detection

Blood was collected from the mice and allowed to stand for 1 h. After centrifugation at 3000 *g* for 15 min, the serum was transferred to a new tube and analyzed using an automatic biochemical analyzer (Mindray, BS-360S).

## Histology

After euthanasia by carbon dioxide, mice were injected with ice-cold PBS through the right ventriculus dexter to remove blood from lung tissue. One left lung lobe was collected and fixed with 4% paraformaldehyde for 24 h. After dehydration by graded ethanol (paraffin section), lung tissues were embedded in paraffin, excised into 5 μm sections, and placed on adhesion microscope slides. After deparaffinization and dehydration, the samples were subjected to

hematoxylin and eosin (H&E) (Beyotime Biotechnology, C0105), alcian blue periodic acid Schiff (AB-PAS) (Solarbio, G1285), and Masson's trichome (Nanjing Jiancheng Bioengineering Institute, D026-1-3) staining according to the manufacturer's protocols. For immunofluorescence analysis, paraffin sections of lung tissue were deparaffinized using graded ethanol, followed by antigen retrieval using ethylenediaminetetraacetic acid, after which the sections were stained with CUL5 and F4/80 antibodies overnight. Images were captured using a BX53 upright fluorescence microscope (Olympus Corporation, Japan) with a 20× dry objective to scan the whole section.

PAS scoring was modified based on previously described methods[32]. Briefly, each airway in one mouse slice was scored based on the following parameters: no visible hyperplasia or mucus production, 0; patchy hyperplasia and/or PAS staining in 0–25%, 1; patchy hyperplasia and/or PAS staining in 25–50% of bronchioles, 2; patchy hyperplasia and/or PAS staining in 50–75%, 3; and patchy hyperplasia and/or PAS staining in 75–100% and/or bronchiolar plugging or obliteration, 4. The reported scores represent the average scores in the airways of each lung calculated as the total score of every airway divided by the total number of airways. H&E scoring was performed using the Aperio Image Scope software. The number of infiltrating cells was calculated based on the number of nuclei. Collagen deposition around the airways were quantified as previous studies[63,64]. Firstly, suitable airways were selected, which were complete, of an appropriate size, not attached to other airways and cut in a plane perpendicular to their length (the minimum internal diameter: maximum internal diameter ratio was more than 0.5 in all cases). Then, the area of collagen in the airway wall (defined as the area between the epithelial basement membrane and airway adventitia) was measured by ImageJ software (National Institutes of Health, USA). The results were expressed as airway wall collagen area (μm$^2$) (blue) divided by the airway circumference (μm). The mean percentage of collagen deposition area in five different regions of each tissue sample was calculated for each mouse. The lung tissue damage score was modified based on previously described methods[65]. The whole lung tissue of mice in paraffin blocks was scored based on the following parameters: no obvious color change, 1; and dark brown discoloration of lung tissue in the paraffin block, 2.

## Immunofluorescence

As described in a previous study[66], cells were fixed with methanol for 20 min at –20 °C and permeabilized with 0.3% H$_2$O$_2$ for 30 min. Paraffin sections of lung tissue were heated in citrate buffer to repair the antigen. The slides were allowed to cool at room temperature. Subsequently, the slides or cells were blocked with 5% normal goat serum (NGS) and 0.1% Tween-20 in tris buffered saline (TBS) for 30 min at room temperature. The samples were then incubated with primary antibodies overnight at 4 °C. After washing thrice with TBS, the cells were further incubated with secondary antibodies (Alexa Fluor 488-conjugated anti-mouse antibody and 594-conjugated anti-rabbit antibody, 1:500). The cell nuclei were stained with 4′,6-diamidino-2-phenylindole (DAPI) (BioLegend, 422801, 100 ng/ml). MitoTracker Deep Red FM (Cell Signaling Technology, 8778P) was used to stain mitochondria. Images were captured using a laser scanning confocal fluorescence microscope (Leica TCS SP8, Leica Microsystems, Wetzlar, Germany). Images were exported and analyzed using the ImageJ software (National Institute of Mental Health, Bethesda, MD, USA). In the co-location analysis, quantization was performed by Pearson's correlation coefficients (PCC) using the ImageJ software (National Institutes of Health) and calculated as all pixels above the background (no thresholds applied), as described in a previous study[67].

## Cell culture

BMDMs were isolated from mouse femurs and tibias as described in a previous study[20]. Briefly, the cells were cultured in Dulbecco's

modified Eagle's medium (DMEM, Gibco, USA) with 10 ng/ml recombinant mouse M-CSF for 6 days. For neutrophil extraction, bone marrow cells were isolated from mouse femurs and tibias, and red blood cells were lysed. The cells were then resuspended in HBSS containing 2% FBS and 2 mM EDTA and added to 3 ml Histopaque 10771 (density, 1.077 g/ml, Sigma, 10771-100 ML) and 3 ml 72% Percoll (Merck, P1644) in a 15 ml conical tube. The tube was centrifuged for 30 min at 750 g at 25 °C without brake. Neutrophils were isolated from the interface between the 72% Percoll and Histopaque 10771 layers and cultured in the corresponding medium. The purity of neutrophils is shown as the percentage of Ly6G positive cells, determined by flow cytometry. The human cervical carcinoma cell lines HeLa (CCL-2) and HEK293T (CRL-11268) were purchased from ATCC and cultured in Roswell Park Memorial Institute (RPMI) medium (RPMI-1640, Gibco) or DMEM supplemented with 10% FCS (Gibco/ThermoFisher) and 1% penicillin/streptomycin (Yeasen Biotech). The human leukemia monocytic cell THP-1 stable cell line (TIB-202) expressing a CUL5 specific shRNA (THP-1-CUL5sh) were generated from our pervious study[30]. THP-1 cells were differentiated into macrophages by incubation in the presence of phorbol-12-myristate-13-acetate (PMA) (100 ng/ml) in DMEM for 24 h. All cells were kept in a humidified cell culture incubator at 37 °C, 5% $CO_2$, and 95% humidity. All cells were routinely tested for mycoplasma every month and confirmed to be negative.

### Transwell assay
Purified neutrophils ($6 \times 10^5$) in 200 μl conditioned medium (CM) were seeded into the upper chambers of transwell plate containing a polycarbonate filter (3 μm pores, Corning, UK). CM (500 μl) was added into the lower chambers in a 24-well trans-well plate. To detect neutrophil migration stimulated by IL-17A, 100 ng/ml IL-17A was added to the upper chambers. After 16 h, the translated neutrophils in the lower chambers were imaged using an IX73 inverted fluorescence microscope (Olympus Corporation, Japan). The translated neutrophils on the bottom of the upper chambers were fixed with precooled methanol at 4 °C and stained with crystal violet (Sigma, 548-62-9). After washing, the neutrophils at the bottom of the upper chambers were gently flicked, and the exterior of the bottom of the upper chamber was imaged using a BX53 upright fluorescence microscope (Olympus Corporation, Japan) after drying naturally. Cell counts in the upper chambers were obtained by thresholding for nuclear staining, followed by automated counting. Total cell counts were recorded as the sum of cell counts from the bottom of the upper and lower chambers. The cell migration index was calculated as the total area of migrating cells in the upper and lower chambers of the corresponding group divided by that of the WT CM group.

### Plasmid construction
cDNA for OGT was obtained from the THP-1 cell line using standard PCR techniques. All the cDNAs were inserted into mammalian expression vectors. Plasmids encoding cGAS, STING, RIG-I, MAVS, TBK1, and IRF3 and IFN-β-luciferase reporter plasmid were a gift from Prof. Dapeng Yan (Fudan University, Shanghai, China). The truncated mutant vectors CUL5 ΔCH, CUL5 ΔNEDD, CUL5 ΔCH + NEDD, CUL5 ΔCR1-3, OGT ΔGTF, OGT ΔPEP, OGT-K306R, OGT-K459R, OGT-K547R, OGT-K550R, OGT-K996R, and OGT-K999R were constructed using the KOD-Plus-Mutagenesis Kit (SMK-101, TOYOBO) according to manufacturer's instructions. Mouse *Ogt* short hairpin RNA (shRNA) vector (pGMLV-SB3) was purchased from Genomeditech. Co., Ltd. (Shanghai, China), and the sequences are shown in Supplementary Table 3. Plasmids encoding CUL5, *CUL5* shRNA, SOCS1, SOCS2, SOCS3, and Ub and their mutants are described in our previously published articles[18,20,30]. All plasmids used for transfection were extracted using Plasmid Extraction Mini-kits (Toroivd, PDE-100).

### RNA isolation and qRT-PCR
Total RNA was extracted from lung tissues or cells using TRIzol reagent (Invitrogen, 15596018) according to the manufacturer's instructions. RNA was reverse transcribed using the ReverTra Ace qPCR RT Kit (FSQ-101, TOYOBO, Japan) according to the manufacturer's instructions. The cDNA was subsequently subjected to Real-Time PCR to quantify the transcripts of the target genes and the housekeeping gene *Gapdh* (glyceraldehyde-3-phosphate dehydrogenase) using SYBR Green Real-time PCR Master Mix (TOYOBO, QST-100). A Real-Time qPCR system (Bio-Rad, CFX Opus96) was used to measure SYBR green incorporation. The qPCR primer sequences are listed in Supplementary Table 3.

### Transfection and luciferase assay
Cells were transiently transfected with polyethyleneimine (Polyscience, Warrington, PA, USA) or Lipofectamine 2000 (11668; Invitrogen). The promoter activities were analyzed using a Dual-Luciferase Reporter Assay Kit (Promega), using the IFN-β luciferase reporter plasmid and the pRL-TK plasmid (*Renilla* luciferase control plasmid, Promega, Madison, WI, USA) served as the internal standard for normalization purposes.

### ELISA
BALF was collected from mice and cell-free supernatants were stored at −80 °C until analysis. After dilution with PBS, the concentrations of IL-4, IL-5, IL-13, IFN-β, and serum IgE were determined using commercially available ELISA kits (R&D Systems, or Thermo Fisher Scientific, 88-50460-22 for IgE) according to the manufacturer's instructions. In all experiments, absorbance was detected using a Varioskan Flash multilabel counter (Thermo Fisher Scientific) to determine concentrations based on a standard curve drawn on each plate.

### Immunoprecipitation and immunoblot analysis
For immunoprecipitation assays, cells were homogenized in cold RIPA buffer (Beyotime Biotechnology, P0013J) with a protease inhibitor cocktail (PIC, Sigma-Aldrich, P8340, 1:100). Protein concentrations were determined using bicinchoninic acid (BCA) Kits according to the manufacturer's protocol. Cell lysates were incubated with 20 μl Protein G Agarose beads (Millipore, 16-266) for 1 h to remove nonspecific protein binding. The lysates were then incubated with antibodies and 40 μl Protein G Agarose beads overnight with constant rotation at 4 °C. The protein complex bound to the beads was washed and subjected to immunoblotting. For immunoblotting, the blots were incubated with primary antibodies, further treated with corresponding secondary antibodies, and reacted with BeyoECL Plus (Beyotime Biotechnology, P0018S), followed by multicolor fluorescence and chemiluminescence imaging (Bio-Rad, CFX Opus 96). Densitometric quantification of protein bands in the western blots was performed using the ImageJ software (National Institute of Mental Health).

### Quantification and statistical analysis
Data are presented as means ± s.e.m. of experimental replicates, including the exact value of n and statistical significance. A *p* value < 0.05 was considered statistically significant. The mean values of paired groups were analyzed using Student's *t* test. The means of multiple groups were analyzed using one- or two-way ANOVA, followed by Tukey's test or Sidak's test. All analyses were performed using GraphPad Prism version 9 (GraphPad Software, San Diego, CA, USA). No data points were excluded from the analysis. Data collection and analyses were blinded to the experimental conditions.

### Reporting summary
Further information on research design is available in the Nature Portfolio Reporting Summary linked to this article.

## Data availability

The authors declare that the data supporting the findings of this study are available within the article or from the corresponding author on request. The RNA-seq raw data are deposited in the GEO database under the primary accession number GSE246601. Other data in the figures are deposited in Figshare with DOI number 10.6084/m9.figshare.24531463.  are provided with this paper.

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

## Acknowledgements

This work was supported by National Key Research and Development Program of China (2023YFA0913700) (F.Q.), National Natural Science Foundation of China 82273934 (L.S.), 82173821 (F.Q.), 82073858 (L.S.), 82373875 (F.Q.), and 81973329 (F.Q.), Natural Science Foundation of Shanghai (21ZR1432700) (L.S.).

## Author contributions

L.S. and F.Q. conceived the study. H.Z., L.S., and F.Q. designed the experiments and drafted the manuscript. X.S. provided the virus and designed the animal study. H.Z. K.X, W.L, X.Y., and Y.G. performed and interpreted experimental data. All authors read and approved the final manuscript.

## Competing interests

The authors declare no competing interests.
