## [Peer Review File · Nature Communications]

Cullin5 drives experimental asthma exacerbations by modulating alveolar macrophage antiviral immunityREVIEWER COMMENTS

Reviewer #1 (Remarks to the Author):

Review of NCOMMS-23-16486

In this manuscript, the authors demonstrate that CUL5 modulates neutrophilic inflammation in a murine model of a virus-induced asthma exacerbation. Specifically, they demonstrate that myeloid cell CUL5 deficiency reduces neutrophilic lung inflammation by augmenting IFN- β production. They then show that TSLP modulates CUL5 expression and that CUL5 induces polyubiquitination of OGT, blocking the effect of OGT on MAVS and RIG-I signaling activation. The authors suggest that CUL5 could be a therapeutic target in virus-induced asthma exacerbations. The manuscript is clearly written and provides some novel insights into the mechanism of action of CUL5. The strength of the research is the convincing mechanistic studies demonstrating how CUL5 ultimately regulates IFN- β production following viral infection. However, the link to viral-induced asthma exacerbations is not fully defined and the focus on alveolar macrophages as the key source of CUL5 may not tell the whole story.

Specific points:

- 1) Its not clear to this reviewer if the model used truly represents a virus-induced asthma exacerbation versus a viral pneumonitis (which is clear in the histology) on top of HDM induced allergic airway inflammation. There is clearly enhanced inflammation with the combination, but I don't think you can say the response is unique without comparing the results to PR8 infection alone. Would CUL5 be induced in that situation and would CUL5 deficiency have similar effects? Could this just be a mechanism to suppress IFN- β in response to viral infection alone? These data are important to determine if this is a unique mechanism for viral-induced asthma exacerbations.
- 2) The authors focus on alveolar macrophages for much of their analysis. Although alveolar macs are clearly important in influenza pneumonitis/alveolitis, their role in asthma is less clear. In fact, airway macrophages and inflammatory monocytes may be more relevant to asthma exacerbations in humans.

- 3) LysMCre mice will delete in granulocytes in addition to monocytes and macrophages. Could the effects of the CUL5 deletion be from effects in neutrophils or monocytes? It would be worthwhile to better define all of the cell-types affected by the deletion.
- 4) IFN- β is made by macrophages as well as other cell types in the lung including epithelial cells and plasmacytoid dendritic cells. Is CUL5 active in these cells? Do the results suggest that alveolar macrophages are the primary source of IFN- β ?
- 5) The TSLP data is less convincing. How does TSLP upregulate CUL5 protein so quickly (within 1 hr)? What happens at later time points?
- 6) TSLP is released as an alarmin but remains low at baseline. How does this explain an increased level of CUL5 at baseline prior to viral infection?
- 7) How do the authors explain the effects on IgE levels if IL-4 levels are not affected in their experiments?
- 8) IFN- β has been tested as a therapy for viral-induced exacerbations in asthma with mixed results. These should be discussed.
- 9) What is the evidence that virus-induced exacerbations are always resistant to corticosteroids as stated in line 69 page 4?
- 10) Some relevant correlative data in human tissue or cells would greatly strengthen the manuscript.

Reviewer #2 (Remarks to the Author):

General overview and summary

In order to further elucidate the mechanisms by which respiratory viral infections exacerbate asthma, this group developed a mouse model. They then analyzed both the cellular and molecular changes that occurred in allergen exposed mice lungs after viral challenge. They analyzed tissues for pathological changes as well as for inflammatory responses. They identified potential molecular drivers using RNA-seq.

Results

Fig 1. Amongst the upregulated genes in alveolar macrophages was Cul5, a scaffold protein involved in the assembly of E3 ligases that ubiquitinate substrates to signal various processes including degradation. They also showed Cul5 was upregulated via

immunofluorescence of macrophages from asthmatic mice.

Comments:

Minor-Fig 1 legend has last part incorrectly indicated as J.

Fig 2. To further study the role of Cul5 they created myeloid specific Cul5 deleted mice. This study revealed that the loss of Cul5 alleviated many of the phenotypes associated with asthma exacerbation, implying Cul5 activity was associated with these responses.

Fig 3. In order to more fully investigate the role of Cul5 they used RNA-seq to compare expression levels in macrophages lacking Cul5. They discovered elevated levels of a number of interferon associated genes as well as the IFN- β gene. They concluded that Cul5 functioned by suppressing IFN- β expression in this system.

Fig 4. The RNA-seq analysis also identified proteins in the RIG-I-like receptor signaling pathway. Further investigation revealed Cul5 knockdown caused accumulation of MAVS in the mitochondria and overexpression of Cul5 caused MAVS multimerization. However, overall expression of MAVS was unchanged by the level of Cul5. Since MAVs activation requires PTMs, they examined their data for pathways that are involved in PTM. They found O-linked glycosylation was upregulated in the absence of Cul5 and specifically O-GlcNAcylation of MAVS was upregulated. OGT is known to be involved in O-GlcNAcylation so they examined its role in Cul5 regulation and concluded that Cul5 negatively regulates IFN- β expression by inhibiting OGT-mediated MAVS O-GlcNAcylation.

Fig.5, 6. Here they demonstrate that Cul5 binds and ubiquitinates OGT which results in its degradation.

Comments (Fig. 6):

Minor-Panel I, the flag gel should be quantitated. The corresponding gel in the supplemental data should also be quantitated. It is not obvious to this reader that these gels look different enough to make the claim of SOCS3 specificity. Panel M, it is possible that OGT binding to Cul5 and SOCS3 is not a single complex. Might be a good idea to show that a Cul5 mutant

that cannot bind SOCS proteins is not brought down in such an IP. Panel N, the ubiquitination in the last lanes appears to be similar. If SOCS3 is required for binding you would expect more. Would be good to quantitate the amount of ubiquitination in these lanes and show the average from your replicates.

Fig 7. Identified signaling pathway that induces Cul5-TSLP.

Comments:

Minor-Panel should show quantification of Cul5 and OGT. It appears that the TSLP experiment had lower Cul5 at the start compared to the others. What is the explanation for that?

Fig 8. Showed that IFN-B alleviates virally induced asthma exacerbation.

Overall summary

This is really an excellent study with high quality data, well written and a logical flow for the presentation. It has several major important discoveries associated with it. These include some insightful information regarding the induction of expression of a cullin. Most regulation of cullins has been shown to be at the level of substrate modification and/or induced expression of substrate adaptors. This study clearly shows the cullin being upregulated in response to signaling. I have no major issues with this manuscript and suggest it be accepted after the minor issues I mentioned have been addressed.

Reviewer #3 (Remarks to the Author)

The authors investigate the role of Cullin-5 in the pathogenesis of influenza-induced asthma exacerbations. This is a significant body of work utilising a wide range of high-end and appropriate experimental techniques. The authors establish a mouse model of influenza-induced asthma exacerbation to show that CUL5 is associated with neutrophilic inflammation in asthma, and that in Cul5 deficient mice, neutrophilic inflammation consistent with influenza infection is mitigated along with several other features of an asthma phenotype. The authors then identified a mechanistic pathway by which CUL5

contributes to an asthma exacerbation, and then assessed the efficacy of IFN β as a therapy against influenza-induced asthma exacerbations. This is a well thought out and well-considered manuscript.

Major Comments

1. Was AHR measured in the establishment of your model of influenza-induced asthma exacerbation? This would be a key measure to show that influenza infection is exacerbating your HDM-derived asthma phenotype as this is what is observed clinically. Please provide.
2. The authors have not provided a methodology for the histological assessment of collagen in the lungs. Supplementary Figure 1 mentions that this was done in whole lung tissues, however a more appropriate and relevant assessment would involve measures within proximity to the airways (consistent with the disease in question) and not within in the parenchyma.

Minor Comments

1. Lines 25, 99, 191, 230, 294, 432, 434, 441, 450, 482, 503, 507. The authors should refer to 'asthma exacerbations' in the plural form. Subsequently, grammatical context needs to be updated with these changes ie. is/are.
2. Lines 45-46. The authors switch between nouns and verbs in the description of asthma. Recommend to use nouns ie. wheeze, cough.
3. Lines 81-83. Sentence mentions several studies, however only one has been referenced.
4. Line 94. Grammatical correction. 'CUL5 has emerged as a critical regulator of..'
5. Lines 120-123 and Lines 182-183. In describing findings from RNAseq data, the authors use the protein form to reference targets ie. capitalised. These should be italicized instead.
6. Line 130. There are 4732 DEGs, not 4733.
7. Line 159. Remove 'greatly'
8. Line 160, 286. The authors refer to airways hyperresponsiveness/reactivity (AHR), tissue damping and elasticity. It should be noted that AHR is a measure of the resistance, tissue damping, elasticity and compliance in response to increasing doses of stimuli (such as methacholine). The authors have presented AHR as measured by resistance, tissue damping and elasticity. This should be corrected throughout the manuscript. Compliance should also be mentioned in the text as it has been presented in Figure 2. Additionally, reference should

be made as to the significance and difference between the resistances in Figure 2a (ie. transpulmonary [Rrs] vs central airways resistance [Rn]), which is important on line 402.

9. Line 180. Remove 'remarkably'

10. Lines 115, 192, 207, 223, 280, 398. 'Asthma-exacerbated' is not an adjective. This should be , revised.

11. Line 207. 'macrophages' should be referred to as 'BMDMs'.

12. Line 220. 'Asthmatic mice' is a contradiction in terms since mice cannot get asthma. Consider revising terminology.

13. Line 443. Revise sentence. 'The type I interferon...'

14. Line 444. Revise sentence. 'An asthma exacerbation is induced.. ..defences..'

15. Line 448. Revise. ILC2s.

16. Line 459. Would it not be more accurate to say that O'GlcNAcylation contributes to several diseases?

17. Line 463. Replace 'it' with 'them'.

18. Line 464. The acronym 'HBP' is not used hereafter, and can therefore be removed.

19. Line 499. It is counter-intuitive to state that an asthma exacerbation is 'regulated'. Consider removing 'the regulation of' from this sentence.

20. Line 512. Please mention the genetic background of the GM mice.

21. Line 522. Please check details of influenza strain. Typically, the last number refers to the year that the strain was isolated. Most A/PR8 strains originate from an isolate from 1934. ie. A/PR8/34.

22. Line 589. Please provide additional details of the HDM used. Was protein content derived by BCA or Bradford Assay? Additionally, XPB82D3A25 is the product number, and while important to include, the Lot Number should also be included (6-digit number). This can be found on the Certificate of Analysis provided by Greer.

23. Line 597. What was the solvent vehicle used?

24. Line 599. Revise. 'Treatment' for 'Infection'.

25. Line 618. What volume of methacholine was nebulised into the mice?

26. Line 636. Mention the purpose of injecting PBS into the mouse. To perfuse the lungs and remove RBCs from the lumen?

27. Line 749. What is meant by 'indicated time'?

28. Supplementary Figure 1 (and throughout). The authors refer to 'periodic acid-Schiff

(PAS) score'. A more accurate terminology would be 'Score of mucus producing cells in the airways' or 'Mucus producing cell score'.

29. Supplementary Figure 3c. Please adjust significance bar to reflect comparison.

30. Supplementary Figures 3c, 3g, 4e-g, 4i-j, 7b. Statistical comparisons should be drawn between appropriate control groups.

Reviewer #4 Joint review with reviewer #3 so no additional comments to authors.

Dear reviewers,

We really appreciate for the comments on our manuscript entitled "Cullin5 drives asthma exacerbation by modulating alveolar macrophage antiviral immunity" (NCOMMS-23-16486A), and the opportunity to resubmit it. In response to the insightful and constructive comments, we have carried out additional experiments. The main revisions are highlighted in yellow in the revised manuscript and the responses to the reviewers' comments are described as below:

Comments from reviewers:

Reviewer #1 (Remarks to the Author):

In this manuscript, the authors demonstrate that CUL5 modulates neutrophilic inflammation in a murine model of a virus-induced asthma exacerbation. Specifically, they demonstrate that myeloid cell CUL5 deficiency reduces neutrophilic lung inflammation by augmenting IFN- β production. They then show that TSLP modulates CUL5 expression and that CUL5 induces polyubiquitination of OGT, blocking the effect of OGT on MAVS and RIG-I signaling activation. The authors suggest that CUL5 could be a therapeutic target in virus-induced asthma exacerbations. The manuscript is clearly written and provides some novel insights into the mechanism of action of CUL5. The strength of the research is the convincing mechanistic studies demonstrating how CUL5 ultimately regulates IFN- β production following viral infection. However, the link to viral-induced asthma exacerbations is not fully defined and the focus on alveolar macrophages as the key source of CUL5 may not tell the whole story.

Specific points:

1) It's not clear to this reviewer if the model used truly represents a virus-induced asthma exacerbation versus a viral pneumonitis (which is clear in the histology) on top of HDM induced allergic airway inflammation. There is clearly enhanced inflammation with the combination, but I don't think you can say the response is unique without comparing the results to PR8 infection alone. Would CUL5 be induced in that situation and would CUL5 deficiency have similar effects? Could this just be a mechanism to suppress IFN- β in response to viral infection alone? These data are important to determine if this is a unique mechanism for viral-induced asthma exacerbations.

Response:

We appreciated the reviewer's comment. The concerns you raised are very important. While many literature sources have utilized the combination of influenza virus and OVA/HDM to induce asthma exacerbation ^{1, 2, 3}, we still thoroughly determined the relevance of the animal model to the clinical asthma exacerbations. We conducted a comparative analysis between HDM+PR8 treatment and PR8 infection alone, to identify any distinctions. As shown in revised Fig. 1, HDM+PR8 treatment group not only showed more inflammatory cell infiltration, serum albumin (ALB) concentration, total cell count and neutrophils infiltration in the BALF compared with PR8 group, but also showed significant AHR, mucus deposition, collagen accumulation, and IgE production compared to simple PR8 infection (revised Fig. 1). These results indicate that HDM+PR8 treatment is a model of virus-induced asthma exacerbation rather than a model of viral pneumonitis.

In addition, we also detected the expression and role of CUL5 in PR8 induced viral pneumonitis mice model. Our data showed that although CUL5 deficiency could decreased the virus content in lung tissue, alleviate PR8-induced lung inflammation, neutrophil infiltration and IFN- β expression in BALF (response Fig. 1a-f), PR8 infection had no effect on the expression of CUL5 in lung tissue (response Fig. 1g). These results suggest that although CUL5 can participate in regulating viral infections, its regulatory mechanism is unique for virus-induced asthma exacerbations due to the induction of CUL5 expression by the specific alarmin molecule TSLP in asthma.

Response Fig. 1 Myeloid-specific CUL5 deficiency alleviates PR8-induced pneumonia. a, Schemes showing the establishment of PR8-induced pneumonia. **b,** H&E-stained lung tissues of the indicated mice. Scale bars, 2mm, 50 μ m. **c,** Relative influenza virus content in the lung tissues of the indicated groups was quantified based on M protein levels measured by RT-qPCR and shown as Δ Ct fold change. **d,** Total cell count in the BALF of the indicated mouse group. **e,** Neutrophil (Ly6G⁺) percentage in BALF was determined. **f,** ELISA was performed to determine IFN- β level in the BALF of the indicated mice. **g,** Immunoblotting of CUL5 and β -

actin in lysates of lung tissue for the indicated group. Data shown are representative of three independent experiments (mean \pm s.e.m.), n = 5 per group per experiment. *p*-values in (c) were calculated by unpaired two-tailed *t*-test, or two-way ANOVA (Tukey's test) in (d)-(f).

2) The authors focus on alveolar macrophages for much of their analysis. Although alveolar macs are clearly important in influenza pneumonitis/alveolitis, their role in asthma is less clear. In fact, airway macrophages and inflammatory monocytes may be more relevant to asthma exacerbations in humans.

Response:

We appreciated the reviewer's comment. Alveolar macrophages are the most abundant innate immune cells in the distal lung, located on the luminal surface of the alveolar space. Although they are usually considered to play an important role in host defense against respiratory pathogens, similar to other tissue-resident macrophages, AMs also perform tissue-specific homeostatic functions in many chronic respiratory diseases^{4,5}.

We have focused on the role of alveolar macrophage in asthma for many years. Many published literatures and our studies have showed that alveolar macrophage is involved in complex immune regulatory reactions with eosinophils, neutrophils, T lymphocytes, mast cells and epithelial cells during the development of asthma^{6,7,8}. For instance, in response to the stimulation of IL-4, IL-13, or TSLP, AMs is required for tissue remodeling and secrete GM-CSF, which recruit eosinophils and promote Th2 differentiation in asthma^{7,9}. In response to TNF- α and IFN- γ stimulation, AMs functioned as a powerful inflammatory regulator and secreted IL-6, TNF- α , and IL-8, which promote neutrophil accumulation, Th17 development, and IL-17A, TGF- β , and

MMP9 expression promoting the secretion of mucus in goblet cells and tissue remodeling in lung tissue ^{10, 11}. Besides, AMs can secrete IFN- β to promote antiviral immune response and neutrophil functions, and inhibit asthma exacerbation ¹¹. Collectively, AMs play an important role in asthma. Due to their long-life span, strong plasticity and functional diversity, we believe that AMs can be used as biomarkers and drug targets for the treatment of asthma. We added the background of the function of AMs in asthma exacerbation in revised manuscript page 4, lines 72-78).

As mentioned by the reviewer, other pulmonary macrophage such as pulmonary interstitial macrophages (IMs) and infiltrating monocytes may also play important role in the pathogenesis of asthma exacerbations, so we detected the expression of CUL5 in these macrophages. We found that, although the expression of CUL5 was also upregulated in IMs (CD45⁺ CD64⁺ CD11c⁻ CD11b⁺ Siglec-F⁻) and monocytes (CD45⁺ CD11b^{med} F4/80^{med}) of virus-induced asthma exacerbation mice compared to control mice, the relative level was lower in comparison with AMs (CD45⁺ CD64⁺ CD11C⁺ Siglec-F⁺) (response Fig. 2). Besides, many evidences have showed that pulmonary interstitial macrophages and infiltrating monocytes can be transformed into alveolar macrophages during inflammatory pathology ^{12, 13}. Thus, our data indicated that AMs is the main source of CUL5 in virus-induced asthmatic exacerbation mice and the regulatory role of CUL5 in virus-induced asthma exacerbation may primarily be mediated through AMs. The function of CUL5 in the other two types of macrophages would be determined in the future.

Response Fig. 2 CUL5 was upregulated in different macrophage subsets of asthma exacerbation mice. **a**, Sorting strategy of alveolar macrophages (AMs, CD45⁺ CD64⁺ CD11c⁺ Siglec-F⁺), Interstitial macrophages (IMs, CD45⁺ CD64⁺ CD11c⁻ CD11b⁺ Siglec-F⁻), monocytes (Monos, CD45⁺ CD11b^{med} F4/80^{med}) in the bronchoalveolar lavage fluid (BALF) of mice by flow cytometry. **b**. *Cul5* mRNA level in indicated cells groups were measured by qPCR. Data represent the mean \pm s.e.m., n = 3 per group. *p*-values were calculated by unpaired two-tailed *t*-test.

3) *LysM^{Cre}* mice will delete in granulocytes in addition to monocytes and macrophages. Could the effects of the *CUL5* deletion be from effects in neutrophils or monocytes? It would be worthwhile to better define all of the cell-types affected by the deletion.

Response:

We appreciated the reviewer's comment. Per the request of the reviewer, we isolated bone marrow derived neutrophils and monocytes to determine the effects of *CUL5* on these cells. As shown in response Fig. 3a, Poly(I:C) stimulation could not induce the expression of IFN- β in both WT neutrophils and *CUL5* deficient neutrophils. *CUL5*

deficiency also had no effect on the migration ability of neutrophils under IL-17 and IFN- β stimulation (response Fig. 3b-c). As shown in response Fig. 3d, CUL5 deficiency could increase Poly(I:C)-induced expression of IFN- β in monocytes, however, the overall expression level of IFN- β in monocytes is much lower than that in macrophages (revised Supplementary Fig. 5c vs response Fig. 3d, more than four hundred-fold in BMDMs vs less than ten hundred-fold in monocytes). Thus, our data suggests that the role of CUL5 in driving virus-induced asthma exacerbation is primarily mediated through macrophages rather than neutrophils and monocytes.

Response Fig. 3 CUL5 deficiency has no influence on the function of neutrophils, and CUL5 weakly induces the expression of IFN- β in monocytes. **a**, IFN- β expression in neutrophils treated with 4 μ g/ml Poly(I:C) for 6 h were measured by qPCR. **b**, Transwell assay analysis of neutrophil migration ability. **c**, The cell migration index represents the total area of migrating cells in the upper and lower chambers of the corresponding group divided by that of the WT PBS group. Scale bar, 100 μ m. Monocyte were separated from bone marrow of mice.

d. IFN- β expression in monocytes treated with 4 $\mu\text{g/ml}$ Poly(I:C) for 6 h were measured by qPCR. Data shown are representative of three independent experiments (b), or mean \pm s.e.m., $n = 3$ biological replicates. p -values were calculated by two-way ANOVA (Tukey's test).

4) IFN- β is made by macrophages as well as other cell types in the lung including epithelial cells and plasmacytoid dendritic cells. Is CUL5 active in these cells? Do the results suggest that alveolar macrophages are the primary source of IFN- β ?

Response:

We appreciated the reviewer's comments. Indeed, PR8 infection can induce IFN- β production in various cells, such as lung epithelial cells and plasmacytoid dendritic cells. However, our study focused on myeloid cell-specific conditional knockout mice, which did not affect lung epithelial cells and plasmacytoid dendritic cells. By constructing $\text{LysM}^{\text{Cre}} \text{Cul5}^{\text{fl/fl}}$ mice, we demonstrated that IFN- β derived from macrophages in myeloid cell-specific Cul5 knockout mice could significantly inhibited the infiltration of neutrophils and lung tissue pathology in virus-induced asthmatic exacerbation mice. Our results suggest that alveolar macrophages are one of the primary sources of IFN- β in virus-induced asthma exacerbation. And our findings proved that macrophage-derived IFN- β is sufficient to influence virus-induced exacerbation of asthma. Of course, we cannot exclude the possibility that CUL5 can affect the pathogenesis of asthma exacerbation by regulating the expression of IFN- β from other cell sources, which would be investigated using other specific CUL5 conditional knockout mice.

5) The TSLP data is less convincing. How does TSLP upregulate CUL5 protein so

quickly (within 1 hr)? What happens at later time points?

Response:

Thanks for the reviewer's comment. Thymic stromal lymphopoietin (TSLP) is an alarmin cytokine that play important role in asthma and many other diseases. It targets on multiple cell lineages to drive several gene expression. TSLP signals through the activation of intracellular JAK/STAT signaling. Accumulating studies have showed that the phosphorylation of STAT5 can be detected within 15 min after TSLP stimulation. Generally, the retardation time from transcription to translation is only 20 min ¹⁴, so the induction of downstream protein expression by TSLP is also very rapid. For example, the expression of anti-apoptosis protein Bcl-XL in bone marrow-derived basophils (BMBas) and BMDCs was significantly upregulated after TSLP treatment for 15 min ¹⁵. TSLP also promoted the expression of senescence protein p16 and p21 in lung epithelial cells in a short time ¹⁶. In our study, we found that TSLP can quickly induce the expression of CUL5, indicating that TSLP is an efficient cytokine with a strong ability to activate macrophage within a short time. As protein expression is a dynamic process, according to the reviewer's suggestion, we detected the long-term expression pattern of CUL5 induced by TSLP. As showed in revised supplementary Fig. 12d, CUL5 was induced at 1 hour after TSLP treatment and reached the peak point at 12 hours. Collectively, our data indicates that TSLP can induce the expression of CUL5 in a short time and lasting for 24 hours.

6) TSLP is released as an alarmin but remains low at baseline. How does this explain

an increased level of CUL5 at baseline prior to viral infection?

Response:

We appreciated the reviewer's comment. In normal lung tissues, the basal expression of TSLP is indeed very low. However, in allergic asthma, TSLP was significantly upregulated and contributes to airway remodeling. In our study, the mRNA level of TSLP in lung was upregulated starting from HDM induced allergic asthmatic inflammation and sustained an increasing trend during asthma exacerbation (revised Fig. 1i). The result of immunofluorescence showed that the induction of CUL5 protein in alveolar macrophages was started from the allergic asthma period and sustained a high-level during asthma exacerbation (revised Fig. 1l). Therefore, our data indicates that the increased expression of TSLP during the early allergic asthma stage can induce the expression of CUL5, and this phenomenon indeed occurs before viral infection.

7) How do the authors explain the effects on IgE levels if IL-4 levels are not affected in their experiments?

Response:

We appreciated the reviewer's comment. In asthma, interleukins released by activated Th2 cells (e.g., IL-4 and IL-13) can activate B cells to enhance the production of immunoglobulin E (IgE). Here, as showed in our revised Fig. 1i, HDM could significantly increase the mRNA level of IL-4 and IL-13 mice. However, further virus infection did not intensity the mRNA level of IL-13, only enhancing IL-4 mRNA level as well as IgE (revised Fig. 1e and 1i), indicating that IL-4, not IL-13, is the main

contributor to enhance IgE expression in virus-induced asthma exacerbation. In addition, although the mRNA level of IL-13 didn't show a significant change in virus-induced asthma exacerbation *Cul5^{fl/fl}* and *LysM^{Cre} Cul5^{fl/fl}* mice, the level of IL-4 were significantly decreased in asthma exacerbation *LysM^{Cre} Cul5^{fl/fl}* mice compared to that in *Cul5^{fl/fl}* mice (revised Fig. 2i). Collectively, our data indicates that CUL5 deficiency can reduce the expression of IL-4, thereby restricting the expression of IgE.

8) IFN- β has been tested as a therapy for viral-induced exacerbations in asthma with mixed results. These should be discussed.

Response:

We appreciated the reviewer's suggestions and we have added the discussion of the mixed results of IFN- β treatment in virus-induced asthma exacerbation in our revised discussion (Page 21-22, lines 457-467). Briefly, IFN- β treatment for virus-induced asthma exacerbation were complicated¹¹. For example, Mansouri et al. demonstrated that treating mice with intranasal IFN- β , which was given the third day after the last HDM treatment, alleviated asthma symptoms in HDM mouse models¹⁷. A recent clinical trial showed that IFN- β treatment within 24 hours, once developed a defined cold/flu, significantly enhanced morning peak expiratory flow recovery, reduced the need for additional treatment, and boosted innate immunity as assessed by blood and sputum biomarkers in difficult-to-treat asthma patients¹⁸. Other clinical trials also emphasize the antiviral type I IFN is more sensitive in asthma patients with neutrophilic airway inflammation and in those prescribed high doses of inhaled corticosteroids¹⁹.

Here, we treat asthma exacerbation mice with IFN- β the third day after HDM treatment and within 24 hours after influenza virus infection, We found that IFN- β treatment has a significant therapeutic effect especially on in acetyl choline associated bronchial hyperresponsiveness, neutrophil infiltration, and neutrophilic inflammatory cytokines as well as IgE production. Collectively, the therapeutic effect of IFN- β on asthma patients is influenced by the timing of administration and pathological characteristics of subjects and need to be further studied.

9) What is the evidence that virus-induced exacerbations are always resistant to corticosteroids as stated in line 69 page 4?

Response:

We appreciated the reviewer's comments and sincerely apologize for not including the references in our original manuscript. We have now added the references in the revised manuscript (Line 69 Page 4). Increasing clinical and experimental evidence strongly implicates that virus-induced exacerbations are always resistant to corticosteroids. For instance, corticosteroids inhibit the clearance of viral particles and suppress the immune response against the virus, which potentially prolong the duration of viral replication and exacerbate the symptoms associated with viral infections ²⁰. In addition, as neutrophils are a component of inflammation in asthma, particularly in patients with viral exacerbations. Asthma with neutrophilic inflammation poorly responds to corticosteroids treatments due to the prolong neutrophil survival and inhibition of neutrophils apoptosis by corticosteroids ^{21, 22}. In our study, the virus content, cell

number and percent of neutrophils in BALF didn't influence by corticosteroids treatment, which suggested the virus-induced asthma exacerbations mice are resistant to corticosteroids treatment (revised Fig. 8). All in all, these reports indicates that corticosteroids are not an effective treatment method for virus-induced asthma exacerbation.

10) Some relevant correlative data in human tissue or cells would greatly strengthen the manuscript.

Response:

Thanks for the reviewer's suggestions. We detected the effect CUL5 on human macrophage by using PMA-induced THP-1 macrophage stable cell line expressing a CUL5 specific shRNA (THP-1-CUL5sh)²³. As showed in the revised supplemental Fig. 5e and supplemental Fig. 6b-c, CUL5 deficient THP-1 cells showed a significant upregulated IFN- β expression as well as TBK1 and IRF3 phosphorylation compared to control cells (THP-1-Scrsh). We believe that these additional data can strengthen our manuscript.

Reviewer #2 (Remarks to the Author):

General overview and summary

In order to further elucidate the mechanisms by which respiratory viral infections exacerbate asthma, this group developed a mouse model. They then analyzed both the cellular and molecular changes that occurred in allergen exposed mice lungs after viral

challenge. They analyzed tissues for pathological changes as well as for inflammatory responses. They identified potential molecular drivers using RNA-seq.

Results

Fig 1. Amongst the upregulated genes in alveolar macrophages was Cul5, a scaffold protein involved in the assembly of E3 ligases that ubiquitinate substrates to signal various processes including degradation. They also showed Cul5 was upregulated via immunofluorescence of macrophages from asthmatic mice.

Comments:

Minor-Fig 1 legend has last part incorrectly indicated as J.

Response:

We appreciated the reviewer's reminder. We corrected the figure legend in the revised Fig. 1.

Fig 2. To further study the role of Cul5 they created myeloid specific Cul5 deleted mice. This study revealed that the loss of Cul5 alleviated many of the phenotypes associated with asthma exacerbation, implying Cul5 activity was associated with these responses.

Fig 3. In order to more fully investigate the role of Cul5 they used RNA-seq to compare expression levels in macrophages lacking Cul5. They discovered elevated levels of a number of interferons associated genes as well as the IFN-B gene. They concluded that

Cul5 functioned by suppressing IFN- β expression in this system.

Fig 4. The RNA-seq analysis also identified proteins in the RIG-I-like receptor signaling pathway. Further investigation revealed Cul5 knockdown caused accumulation of MAVS in the mitochondria and overexpression of Cul5 caused MAVS multimerization. However, overall expression of MAVS was unchanged by the level of Cul5. Since MAVS activation requires PTMs, they examined their data for pathways that are involved in PTM. They found O-linked glycosylation was upregulated in the absence of Cul5 and specifically O-GlcNAcylation of MAVS was upregulated. OGT is known to be involved in O-GlcNAcylation so they examined its role in Cul5 regulation and concluded that Cul5 negatively regulates IFN- β expression by inhibiting OGT-mediated MAVS O-GlcNAcylation.

Fig.5, 6. Here they demonstrate that Cul5 binds and ubiquitinates OGT which results in its degradation.

Comments (Fig. 6):

Minor-Panel I, the flag gel should be quantitated. The corresponding gel in the supplemental data should also be quantitated. It is not obvious to this reader that these gels look different enough to make the claim of SOCS3 specificity. Panel M, it is possible that OGT binding to Cul5 and SOCS3 is not a single complex. Might be a good idea to show that a Cul5 mutant that cannot bind SOCS proteins is not brought down in

such an IP. Panel N, the ubiquitination in the last lanes appears to be similar. If SOCS3 is required for binding you would expect more. Would be good to quantitate the amount of ubiquitination in these lanes and show the average from your replicates.

Response:

We appreciated the reviewer's comment and have made revisions to Figure 6.

For Panel I, we had quantitated the Flag gel, and the quantitative data were showed in revised supplemental Fig. 11c. Our data showed that although overexpression of SOCS1, 2, and 3 could to some extent decrease the protein levels of OGT, only the downregulation of OGT mediated by SOCS3 can be reversed after the addition of MG-132, suggesting that the SOCS3 is specifically involved in the reduction of OGT protein mediated by proteasome.

CUL5 exerts its function through the forming the Cullin-RING ligase 5 (CRL5) complex. Structurally, CUL5 interacts with the Elongin B/C complex, a RING finger protein (RBX2), and a SOCS protein to form the CRL5 E3 ubiquitin ligase protein complex. The N-terminal region of CUL5, which contains three cullin repeats (CR1 to CR3), is crucial for its binding with SOCS proteins^{24,25}. To address reviewer's comment regarding the interaction between CUL5, SOCS3, and OGT in Panel M, we performed the experiments to assess the interactions among these proteins upon mutating the binding region of CUL5 (CUL5 Δ CR1-3) with SOCS3. As showed in response Fig. 4, CUL5 mutant (CUL5 Δ CR1-3) was still able to contact with OGT, however SOCS3 lost its ability to increase the binding of CUL5 with OGT. This result suggested that CUL5, OGT, and SOCS3 can form a complex, the interaction of CUL5 and OGT is not

dependent on SOCS3, but the effect of SOCS3 on promoting OGT ubiquitination is dependent on the interaction of SOCS3 with CUL5.

For Panel n, we had replaced the image with a shorter exposure, making the ubiquitination of OGT more prominent. We also qualified the amount of ubiquitinated OGT protein of replicated experiments, and showed the results in revised Supplementary Fig. 11d.

Response Fig. 4 SOCS3 promotes the interaction of OGT and CUL5 through binding with CUL5. Co-immunoprecipitation (Co-IP) and immunoblot analyses of HEK293T cells co-transfected with Flag-OGT, Myc-CUL5 Δ CR1-3 mutant, and Myc-SOCS3. Data are representative of three independent experiments.

Fig 7. Identified signaling pathway that induces Cul5-TSLP.

Comments:

Minor-Panel should show quantification of Cul5 and OGT. It appears that the TSLP experiment had lower Cul5 at the start compared to the others. What is the explanation for that?

Response:

We appreciated the reviewer's comment and added the quantification data in revised supplementary Fig. 12c. For western blot experiments, we know that the depth of the bands is affected by the length of exposure time. In our experiment, we focused on the change of CUL5 expression under different stimuli, and did not compare the basal level of CUL5. In order not to cause ambiguity, we replaced some images in Fig. 7 with shorter exposure time from another repeated experiment and carried out quantitative analysis on the three repeated experiments. The qualified result in revised supplemental Fig. 12c showed that the basal level of CUL5 were not of significant difference.

Fig 8. Showed that IFN-B alleviates virally induced asthma exacerbation.

Overall summary

This is really an excellent study with high quality data, well written and a logical flow for the presentation. It has several major important discoveries associated with it. These include some insightful information regarding the induction of expression of a cullin. Most regulation of cullins has been shown to be at the level of substrate modification and/or induced expression of substrate adaptors. This study clearly shows the cullin being upregulated in response to signaling. I have no major issues with this manuscript and suggest it be accepted after the minor issues I mentioned have been addressed.

Reviewer #3 (Remarks to the Author):

The authors investigate the role of Cullin-5 in the pathogenesis of influenza-induced

asthma exacerbations. This is a significant body of work utilising a wide range of high-end and appropriate experimental techniques. The authors establish a mouse model of influenza-induced asthma exacerbation to show that CUL5 is associated with neutrophilic inflammation in asthma, and that in Cul5 deficient mice, neutrophilic inflammation consistent with influenza infection is mitigated along with several other features of an asthma phenotype. The authors then identified a mechanistic pathway by which CUL5 contributes to an asthma exacerbation, and then assessed the efficacy of IFN β as a therapy against influenza-induced asthma exacerbations. This is a well thought out and well-considered manuscript.

Major Comments

1. Was AHR measured in the establishment of your model of influenza-induced asthma exacerbation? This would be a key measure to show that influenza infection is exacerbating your HDM-derived asthma phenotype as this is what is observed clinically. Please provide.

Response:

We appreciated the reviewer's suggestions and we have added the results of AHR measurement in our revised Fig. 1b and relative description in revised manuscript (page 6, lines 118-124). The results showed a significant increase in resistance of respiratory system and airway of influenza-induced asthma exacerbation mice. And the respiratory system compliance was significantly decreased. In addition, our data also showed that the tissue damping and elasticity were greatly increased in virus-induced asthma

exacerbation mice compared with allergic asthma mice. Collectively, our data demonstrates that combined with HDM, influenza infection can induce asthma exacerbation with significant AHR that is mimic to clinical disease symptom.

2. The authors have not provided a methodology for the histological assessment of collagen in the lungs. Supplementary Figure 1 mentions that this was done in whole lung tissues, however a more appropriate and relevant assessment would involve measures within proximity to the airways (consistent with the disease in question) and not within in the parenchyma.

Response:

We appreciated the reviewer's comments. As mentioned by the reviewer, abnormal airway collagen deposition in the airway is associated with the pathogenesis and progression of asthma exacerbation. In order to more accurately describe the collagen deposition of the histological assessment, we re-quantified the deposition of collagen proximity to the airway according to a previous study^{26, 27}. The detail methodology for the histological assessment were described in the methods section in our revised manuscript (Page 32). Briefly, suitable airways were firstly selected (defined by the minimum internal diameter: maximum internal diameter ratio was more than 0.5 in all cases). Then we utilized ImageJ software to scan the area of collagen in the airway wall (defined as the area between the epithelial basement membrane and airway adventitia). Relative collage deposition was calculated by dividing the collagen area (blue) by the airway circumference. We believe that the new quantitative method can more accurately

reflect the collagen deposition in the airways during virus-induced exacerbation of asthma, and be more closely related to the pathological changes of the disease.

Minor Comments

1. Lines 25, 99, 191, 230, 294, 432, 434, 441, 450, 482, 503, 507. The authors should refer to ‘asthma exacerbations’ in the plural form. Subsequently, grammatical context needs to be updated with these changes ie. is/are.
2. Lines 45-46. The authors switch between nouns and verbs in the description of asthma. Recommend to use nouns ie. wheeze, cough.
3. Lines 81-83. Sentence mentions several studies, however only one has been referenced.
4. Line 94. Grammatical correction. ‘CUL5 has emerged as a critical regulator of..’
5. Lines 120-123 and Lines 182-183. In describing findings from RNAseq data, the authors use the protein form to reference targets ie. capitalised. These should be italicized instead.
6. Line 130. There are 4732 DEGs, not 4733.
7. Line 159. Remove ‘greatly’

Response:

We appreciated the reviewer’s comments. For minor comments from 1 to 7, we have carefully checked and revised the typographical and grammar errors. All the changes were highlighted in yellow in the revised manuscript.

8. Line 160, 286. The authors refer to airways hyperresponsiveness/reactivity (AHR), tissue damping and elasticity. It should be noted that AHR is a measure of the resistance, tissue damping, elasticity and compliance in response to increasing doses of stimuli (such as methacholine). The authors have presented AHR as measured by resistance, tissue damping and elasticity. This should be corrected throughout the manuscript. Compliance should also be mentioned in the text as it has been presented in Figure 2. Additionally, reference should be made as to the significance and difference between the resistances in Figure 2a (ie. transpulmonary [R_{rs}] vs central airways resistance [R_n]), which is important on line 402.

Response:

We appreciated the reviewer's comments. We have revised the corresponding description in the whole manuscript. As mentioned in our revised results and method section (Line 117-123, 169, 296, and 414), we used SCIREQ pulmonary system to assess AHR in mice through a methacholine challenge tests (MCTs). All the parameters, including respiratory system resistance (R_{rs}), respiratory system compliance (C_{rs}), central airway resistance (R_n), tissue damping (G), and tissue elastance (H), reflected changes of airway hyperreactivity (AHR). In addition, we had described the significance and difference of these parameters, and added the reference in our revised methods section (Page 30, ref 37 and 45). As mentioned in the method, respiratory system resistance (R_{rs}) reflects the dynamic resistance in the lung, central airway resistance (R_n) represents the resistance of the central airways, respiratory system compliance (C_{rs}) provides a characterization of the overall elastic properties that the

respiratory system needs to overcome during tidal breathing to move air in and out of the lungs, tissue damping (G) reflects the energy dissipation in the alveoli, and tissue elastance (H) reflects the energy conservation in the alveoli. We believe the reference can help us to explain that why our data suggest that IFN- β not only has an effect on the airways, but also has a good improvement effect on airway hyperresponsiveness caused by lung tissue damage.

9. Line 180. Remove 'remarkably'

10. Lines 115, 192, 207, 223, 280, 398. 'Asthma-exacerbated' is not an adjective. This should be , revised.

11. Line 207. 'macrophages' should be referred to as 'BMDMs'.

12. Line 220. 'Asthmatic mice' is a contradiction in terms since mice cannot get asthma. Consider revising terminology.

13. Line 443. Revise sentence. 'The type I interferon...'

14. Line 444. Revise sentence. 'An asthma exacerbation is induced.. ..defences..'

15. Line 448. Revise. ILC2s.

16. Line 459. Would it not be more accurate to say that O'GlcNAcylation contributes to several diseases?

17. Line 463. Replace 'it' with 'them'.

18. Line 464. The acronym 'HBP' is not used hereafter, and can therefore be removed.

19. Line 499. It is counter-intuitive to state that an asthma exacerbation is 'regulated'.

Consider removing 'the regulation of' from this sentence.

Response:

We appreciated the reviewer's comments. For minor comments from 9 to 19, we have carefully checked and revised the grammar errors. We also described the sentences more professional and accurate. All the changes were highlighted in yellow in the revised manuscript.

20. Line 512. Please mention the genetic background of the GM mice.

21. Line 522. Please check details of influenza strain. Typically, the last number refers to the year that the strain was isolated. Most A/PR8 strains originate from an isolate from 1934. I.e. A/PR8/34.

22. Line 589. Please provide additional details of the HDM used. Was protein content derived by BCA or Bradford Assay? Additionally, XPB82D3A25 is the product number, and while important to include, the Lot Number should also be included (6-digit number). This can be found on the Certificate of Analysis provided by Greer.

23. Line 597. What was the solvent vehicle used?

Response:

We appreciated the reviewer's comments. For minor comments from 20 to 23, we have added the missing information in our revised manuscript. For the comment 20, both *Cul5^{fl/fl}* mice and *LysM^{Cre}* mice was generated from C57BL/6 mice. For the comment 21, the influenza strain was PR8A/Puerto Rico/8/34. For the comment 22, the details of HDM used in our experiments were added in the revised manuscript (Page 28). The catalog number is XPB82D3A25, and the Lot. Number is 385931. The product we

purchased is in the 25mg specification, and the specific protein content is detailed in the accompanying instruction manual. Following the instructions, we made 500 μ L aliquots of 5 mg/ml HDM in H₂O and stored it at -80 °C for further experiments. For the comment 23, the vehicle solvent composed with volume ratio of PEG300: PBS = 3:7.

24. Line 599. Revise. 'Treatment' for 'Infection'.

25. Line 618. What volume of methacholine was nebulised into the mice?

26. Line 636. Mention the purpose of injecting PBS into the mouse. To perfuse the lungs and remove RBCs from the lumen?

27. Line 749. What is meant by 'indicated time'?

28. Supplementary Figure 1 (and throughout). The authors refer to 'periodic acid-Schiff (PAS) score'. A more accurate terminology would be 'Score of mucus producing cells in the airways' or 'Mucus producing cell score'.

Response:

We appreciated the reviewer's comments. For minor comments from 24 to 28, we described the sentences more professional and accurate. In addition, we also added the missing information as mentioned by the reviewer in our revised manuscript.

29. Supplementary Figure 3c. Please adjust significance bar to reflect comparison.

30. Supplementary Figures 3c, 3g, 4e-g, 4i-j, 7b. Statistical comparisons should be drawn between appropriate control groups.

Response:

We appreciated the reviewer's comments. For minor comments from 29 to 30, we checked all the statistical comparisons of our manuscript, and adjusted the significance bar. In addition, the statistical comparisons were revised and drawn between appropriate control groups.

Reviewer #4 Joint review with reviewer #3 so no additional comments to authors.

References

1. Makino A, *et al.* RSV infection-elicited high MMP-12-producing macrophages exacerbate allergic airway inflammation with neutrophil infiltration. *iScience* **24**, 103201 (2021).
2. Liu X, *et al.* Proteomic Analysis Reveals a Novel Therapeutic Strategy Using Fludarabine for Steroid-Resistant Asthma Exacerbation. *Front Immunol* **13**, 805558 (2022).
3. Vanders RLP, *et al.* Inflammatory and anti-viral responses to influenza A virus infection are dysregulated in pregnant mice with allergic airway disease. *Am J Physiol Lung Cell Mol Physiol*, (2023).
4. Lugg ST, Scott A, Parekh D, Naidu B, Thickett DR. Cigarette smoke exposure and alveolar macrophages: mechanisms for lung disease. *Thorax* **77**, 94-101 (2022).
5. Dong T, *et al.* Mitochondrial metabolism mediated macrophage polarization in chronic lung diseases. *Pharmacol Ther* **239**, 108208 (2022).
6. Woo YD, Jeong D, Chung DH. Development and Functions of Alveolar Macrophages. *Molecules and cells* **44**, 292-300 (2021).
7. Zhang H, *et al.* AMFR drives allergic asthma development by promoting alveolar macrophage-derived GM-CSF production. *J Exp Med* **219**, e20211828 (2022).
8. Sun L, *et al.* Staphylococcal virulence factor HlgB targets the endoplasmic-reticulum-resident E3 ubiquitin ligase AMFR to promote pneumonia. *Nat Microbiol* **8**, 107-120 (2023).
9. Li CH, Tsai ML, Chiou HC, Lin YC, Liao WT, Hung CH. Role of Macrophages in Air Pollution

Exposure Related Asthma. *Int J Mol Sci* **23**, (2022).

10. Allard B, Panariti A, Martin JG. Alveolar Macrophages in the Resolution of Inflammation, Tissue Repair, and Tolerance to Infection. *Front Immunol* **9**, 1777 (2018).
11. Nakagome K, Nagata M. Innate Immune Responses by Respiratory Viruses, Including Rhinovirus, During Asthma Exacerbation. *Front Immunol* **13**, 865973 (2022).
12. Hussell T, Bell TJ. Alveolar macrophages: plasticity in a tissue-specific context. *Nat Rev Immunol* **14**, 81-93 (2014).
13. Li F, Piattini F, Pohlmeier L, Feng Q, Rehrauer H, Kopf M. Monocyte-derived alveolar macrophages autonomously determine severe outcome of respiratory viral infection. *Science immunology* **7**, eabj5761 (2022).
14. Schott J, *et al.* Nascent Ribo-Seq measures ribosomal loading time and reveals kinetic impact on ribosome density. *Nature methods* **18**, 1068-1074 (2021).
15. Lv J, *et al.* Airway epithelial TSLP production of TLR2 drives type 2 immunity in allergic airway inflammation. *Eur J Immunol* **48**, 1838-1850 (2018).
16. Wu J, *et al.* Central role of cellular senescence in TSLP-induced airway remodeling in asthma. *PLoS One* **8**, e77795 (2013).
17. Mansouri S, *et al.* In vivo reprogramming of pathogenic lung TNFR2(+) cDC2s by IFN β inhibits HDM-induced asthma. *Science immunology* **6**, (2021).
18. Djukanović R, *et al.* The effect of inhaled IFN- β on worsening of asthma symptoms caused by viral infections. A randomized trial. *Am J Respir Crit Care Med* **190**, 145-154 (2014).
19. JL S, *et al.* Reduced Antiviral Interferon Production in Poorly Controlled Asthma Is Associated With Neutrophilic Inflammation and High-Dose Inhaled Corticosteroids. *Chest* **149**, 704-713 (2016).
20. Saturni S, Contoli M, Spanevello A, Papi A. Models of Respiratory Infections: Virus-Induced Asthma Exacerbations and Beyond. *Allergy, asthma & immunology research* **7**, 525-533 (2015).
21. "Cellular mechanisms underlying steroid-resistant asthma." Ridhima Wadhwa, Kamal Dua, Ian M. Adcock, Jay C. Horvat, Richard Y. Kim and Philip M. Hansbro. *Eur Respir Rev* 2019; 28: 190021. *European respiratory review : an official journal of the European Respiratory Society* **28**, (2019).
22. Hammad H, Lambrecht BN. The basic immunology of asthma. *Cell* **184**, 1469-1485 (2021).

23. Zhu Z, Sun L, Hao R, Jiang H, Qian F, Ye RD. Ned8 modification of Cullin-5 regulates lipopolysaccharide-induced acute lung injury. *Am J Physiol Lung Cell Mol Physiol* **313**, L104-L114 (2017).
24. Li Z, *et al.* Cullin-5 (CUL5) as a potential prognostic marker in a pan-cancer analysis of human tumors. *Bioengineered* **12**, 5348-5360 (2021).
25. Zhao Y, Xiong X, Sun Y. Cullin-RING Ligase 5: Functional characterization and its role in human cancers. *Seminars in cancer biology* **67**, 61-79 (2020).
26. Yang Y, *et al.* Dynamic evolution of emphysema and airway remodeling in two mouse models of COPD. *BMC pulmonary medicine* **21**, 134 (2021).
27. Reinhardt AK, Bottoms SE, Laurent GJ, McAnulty RJ. Quantification of collagen and proteoglycan deposition in a murine model of airway remodelling. *Respir Res* **6**, 30 (2005).

REVIEWERS' COMMENTS

Reviewer #1 (Remarks to the Author):

The authors have adequately addressed most of my concerns. However, I have a few minor points that could be addressed to improve the manuscript.

1) Regarding Response to point #1: The inclusion of a PR8 infection alone experimental group has strengthened the results. As shown in the rebuttal, deficiency of Cul5 also enhances viral clearance, reduces neutrophil recruitment, and enhances IFN β production in influenza infection alone. The inclusion of this data in the manuscript would provide further proof of the concept that Cul5 is detrimental in viral infections and that lower levels induced by TSLP may lead to more inflammation with viral infections in asthmatics.

2) Regarding Response to point #2: Although the authors provide a well-reasoned response to point #2, they have not fully addressed my point. I do not doubt that alveolar macrophages are important in the model used in the manuscript, but in humans, asthma is largely a disease of the conducting airways, which are poorly modeled in mice. Conducting airways in humans are lined by pseudostratified epithelium and are rich with innate immune cells such as airway macrophages and dendritic cells, while in mice, the airways below the main bronchi are largely simple columnar epithelium. In fact, the lung pathology of the HDM-PR8 mice shown in the paper displays prominent pneumonitis, which is not the case for most viral-induced asthma exacerbations in patients (usually clear chest X-rays). I think this can be addressed in the discussion for the paper.

3) The authors state that alveolar macs are the most abundant innate immune cell in the lung but their data (response Fig 2) and those from others suggest that monocytes are more abundant.

4) Response to point #9: I don't think the statement that virus-induced exacerbations are always resistant to corticosteroids is accurate. Corticosteroids are standard therapy for asthma exacerbations, and the majority of exacerbations are due to viral infections. Clinical experience has shown that the administration of steroids for asthma exacerbations reduces emergency room visits and hospitalizations for asthma.

Reviewer #2 (Remarks to the Author):

I feel that the issues I had with the earlier submission have been fully addressed to my satisfaction and the manuscript is appropriate for publication.

Reviewer #3 (Remarks to the Author):

I commend the authors on the revisions they have made on this manuscript. I have 3 additional corrections;

- 1) Line 605 - 'powder' has been misspelt. A more accurate term would be 'lypholized powder'
- 2) Line 609 – It is unclear how intranasal sensitization was achieved through only one nostril? Is this an error? Consider revising if so.
- 3) Line 1405 - The gene form of Interferon beta should be used as has been for the rest of the manuscript.

Dear reviewers,

We really appreciate for the comments on our manuscript entitled "Cullin5 drives experimental asthma exacerbations by modulating alveolar macrophage antiviral immunity" (NCOMMS-23-16486A), and the opportunity to resubmit it. In response to the insightful and constructive comments, we have modified several descriptions in the revised manuscript. We have also carefully edited our manuscript to fulfill the request of Nature Communications. The main revisions are highlighted in yellow in the revised manuscript and the responses to the reviewers' comments are described as below:

REVIEWERS' COMMENTS

Reviewer #1 (Remarks to the Author):

The authors have adequately addressed most of my concerns. However, I have a few minor points that could be addressed to improve the manuscript.

1) Regarding Response to point #1: The inclusion of a PR8 infection alone experimental group has strengthened the results. As shown in the rebuttal, deficiency of Cul5 also enhances viral clearance, reduces neutrophil recruitment, and enhances IFNbeta production in influenza infection alone. The inclusion of this data in the manuscript would provide further proof of the concept that Cul5 is detrimental in viral infections and that lower levels induced by TSLP may lead to more inflammation with viral infections in asthmatics.

Response:

We appreciated the reviewer's comment. We have added these results in the revised

supplementary figure 4.

2) Regarding Response to point #2: Although the authors provide a well-reasoned response to point #2, they have not fully addressed my point. I do not doubt that alveolar macrophages are important in the model used in the manuscript, but in humans, asthma is largely a disease of the conducting airways, which are poorly modeled in mice. Conducting airways in humans are lined by pseudostratified epithelium and are rich with innate immune cells such as airway macrophages and dendritic cells, while in mice, the airways below the main bronchi are largely simple columnar epithelium. In fact, the lung pathology of the HDM-PR8 mice shown in the paper displays prominent pneumonitis, which is not the case for most viral-induced asthma exacerbations in patients (usually clear chest X-rays). I think this can be addressed in the discussion for the paper.

Response:

We appreciated the reviewer's comment. In our study, we performed an HDM+PR8-induced asthma exacerbation animal model. Although this model has been widely used in asthma research, it is indeed different from human. In human, asthma exacerbations are primarily characterized by bronchi contract, airway swelling, mucus secretion, and a relatively weak inflammatory response. In mice, the lung pathology of the HDM+PR8 mice displayed prominent pneumonitis. However, this does not mean that airway inflammation is not important in human asthma exacerbations. In both human and mice, virus-induced asthma exacerbations are closely associated with airway inflammation,

and the AHR of asthma exacerbations patients are largely caused by airway inflammation¹. Although human airway macrophages, including those located on the luminal surface of the alveolar space and those associated with the epithelium of the conducting airways, have some phenotypic differences from mouse alveolar macrophages, these cells also participate in regulating airway inflammation in asthma exacerbations. In addition, we detected the role of CUL5 in regulating AHR in our asthma exacerbations mice. Therefore, we believe that the animal model we used can simulate human asthma exacerbations responses in terms of airway hyperresponsiveness and airway inflammation to a certain extent, which helping us determine the pathogenesis of the disease and identify new drug targets. Certainly, we still need to build more suitable models for further in-depth research. To provide a more precise description of our study, we have modified the title, adding a word “experimental”, and providing corresponding discussion in the section of discussion (Lines 475-488).

3) The authors state that alveolar macs are the most abundant innate immune cell in the lung but their data (response Fig 2) and those from others suggest that monocytes are more abundant.

Response:

We greatly appreciated the reviewer for pointing out our oversight in writing. We intended to convey that alveolar macrophages are the most abundant innate immune cells in the alveoli, not in the lungs, as shown in Figure 2 in our manuscript, accounting

for over 90%. In lung tissue, there is a much greater variety of immune cells than in the alveoli, including lymphocytes, interstitial macrophages, monocytes, dendritic cells, neutrophils and so on. The data presented in response Figure 2 is from lung tissue, hence the proportion of alveolar macrophages is not very high. We have modified the description in the revised manuscript (Line 73).

4) Response to point #9: I don't think the statement that virus-induced exacerbations are always resistant to corticosteroids is accurate. Corticosteroids are standard therapy for asthma exacerbations, and the majority of exacerbations are due to viral infections. Clinical experience has shown that the administration of steroids for asthma exacerbations reduces emergency room visits and hospitalizations for asthma.

Response:

We appreciated the reviewer's comment that the description of corticosteroids therapy in the manuscript was not accurate enough. Currently, the standard treatment for virus-induced asthma exacerbations primarily relies on adequate doses of inhaled steroids therapy. However, the efficacy of this treatment approach is limited, and the sensitivity to steroid therapy varies among different types of virus-induced asthma exacerbation².³ Moreover, excessive use of steroids can induce immune suppression and other adverse reactions⁴. Therefore, it is necessary for us to explore additional treatment options. We have modified it in the revised manuscript (Lines 47-48).

Reviewer #2 (Remarks to the Author): I feel that the issues I had with the earlier

submission have been fully addressed to my satisfaction and the manuscript is appropriate for publication.

Response:

We are delighted to address the comments of the reviewers and express our gratitude for their positive evaluation of our manuscript.

Reviewer #3 (Remarks to the Author):

I commend the authors on the revisions they have made on this manuscript. I have 3 additional corrections;

1) Line 605 - 'powder' has been misspelt. A more accurate term would be 'lypholized powder'

2) Line 609 – It is unclear how intranasal sensitization was achieved through only one nostril? Is this an error? Consider revising if so.

3) Line 1405 - The gene form of Interferon beta should be used as has been for the rest of the manuscript.

Response:

We appreciated the reviewer's comment. In the revised manuscript, the term 'powder' has been modified to 'lypholized powder'. In the method section, mice were sensitized intranasally using 250 µg/kg HDM in 20 µl PBS, 10 µl per nostril. We have modified these in the revised manuscript (lines 609 and 613). The gene form of Interferon beta expressed from THP-1 cells have been changed to *IFNB* in the revised supplementary figure legend 6.

References:

1. Holt PG, Sly PD. Viral infections and atopy in asthma pathogenesis: new rationales for asthma prevention and treatment. *Nat Med* **18**, 726-735 (2012).
2. Brandwijk R, *et al.* Pitfalls in complement analysis: A systematic literature review of assessing complement activation. *Front Immunol* **13**, 1007102 (2022).
3. Liu X, *et al.* Proteomic Analysis Reveals a Novel Therapeutic Strategy Using Fludarabine for Steroid-Resistant Asthma Exacerbation. *Front Immunol* **13**, 805558 (2022).
4. Ramsahai JM, Hansbro PM, Wark PAB. Mechanisms and Management of Asthma Exacerbations. *Am J Respir Crit Care Med* **199**, 423-432 (2019).